# Cre/*lox* regulated conditional rescue and inactivation with zebrafish UFlip alleles generated by CRISPR-Cas9 targeted integration

Fang Liu[1†], Sekhar Kambakam[1†], Maira P Almeida[1,2†‡], Zhitao Ming[1†], Jordan M Welker[1,2§], Wesley A Wierson[1,3#], Laura E Schultz-Rogers[1,2¶], Stephen C Ekker[4], Karl J Clark[4], Jeffrey J Essner[1,2], Maura McGrail[1,2*]

[1]Department of Genetics, Development and Cell Biology, Iowa State University, Ames, United States; [2]Interdepartmental Graduate Program in Genetics and Genomics, Iowa State University, Ames, United States; [3]Interdepartmental Graduate Program in Molecular, Cellular and Developmental Biology, Iowa State University, Ames, United States; [4]Department of Biochemistry and Molecular Biology, Mayo Clinic, Rochester, United States

*For correspondence:
mmcgrail@iastate.edu

[†]These authors contributed equally to this work

Present address: [‡]Temerty Faculty of Medicine, University of Toronto, Toronto, Canada; [§]Department III – Developmental Genetics, Max Planck Institute for Heart and Lung Research, Bad Nauheim, Germany; [#]LEAH Labs, Rochester, Rochester, United States; [¶]Department of Pathology and Lab Medicine, University of North Carolina, Chapel Hill, North Carolina, United States

**Abstract** The ability to regulate gene activity spatially and temporally is essential to investigate cell-type-specific gene function during development and in postembryonic processes and disease models. The Cre/*lox* system has been widely used for performing cell and tissue-specific conditional analysis of gene function in zebrafish. However, simple and efficient methods for isolation of stable, Cre/*lox* regulated zebrafish alleles are lacking. Here, we applied our GeneWeld CRISPR-Cas9 targeted integration strategy to generate floxed alleles that provide robust conditional inactivation and rescue. A universal targeting vector, UFlip, with sites for cloning short homology arms flanking a floxed 2A-mRFP gene trap, was integrated into an intron in *rbbp4* and *rb1*. *rbbp4*[off] and *rb1*[off] integration alleles resulted in strong mRFP expression, >99% reduction of endogenous gene expression, and recapitulated known indel loss-of-function phenotypes. Introduction of Cre led to stable inversion of the floxed cassette, loss of mRFP expression, and phenotypic rescue. *rbbp4*[on] and *rb1*[on] integration alleles did not cause phenotypes in combination with a loss-of-function mutation. Addition of Cre led to conditional inactivation by stable inversion of the cassette, gene trapping and mRFP expression, and the expected mutant phenotype. Neural progenitor Cre drivers were used for conditional inactivation and phenotypic rescue to showcase how this approach can be used in specific cell populations. Together these results validate a simplified approach for efficient isolation of Cre/*lox*-responsive conditional alleles in zebrafish. Our strategy provides a new toolkit for generating genetic mosaics and represents a significant advance in zebrafish genetics.

## Editor's evaluation

This technical paper describes a novel strategy for conditional mutagenesis in zebrafish. It develops a simple approach to isolate mutagenic Cre/*lox* conditional alleles for tissue-specific gene inactivation. This work will be of great interest for the zebrafish community, advancing the exciting genetic manipulation tool box in this model organism.

## Introduction

The ability to regulate gene activity spatially and temporally is essential to investigate cell-type-specific gene function during development and in postembryonic processes and disease models. Conditional gene inactivation can be achieved with the Cre/lox system in which the bacteriophage Cre recombinase promotes site-specific recombination at compatible loxP sites engineered in a gene of interest (*Sauer and Henderson, 1988*). Effective application of Cre/lox in vertebrates requires the isolation of transgenic lines expressing Cre, and gene modification to introduce loxP sequences at desired locations. Homologous recombination driven site-directed genetic modification in embryonic stem cells enabled creation of the mouse Cre/lox resource that contains more than 2500 Cre drivers and lox conditional gene alleles (*Bult et al., 2019*). In zebrafish, the lack of embryonic stem cell technology and inefficient homologous recombination in the early embryo necessitated alternative approaches to generate Cre/lox tools (*Carney and Mosimann, 2018*). Tol2 transposon transgenesis (*Balciunas et al., 2006*; *Kawakami et al., 2000*) combined with defined promoters (*Mosimann et al., 2011*) or BAC recombineering (*Förster et al., 2017*) has been used to develop Cre lines, and a rich source of zebrafish cell-type-specific Cre drivers has been generated by random Tol2 transposon enhancer trap screens (*Jungke et al., 2015*; *Marquart et al., 2015*; *Tabor et al., 2019*; *Zhong et al., 2019*). Cre/lox regulated conditional control of gene function in zebrafish was previously limited to generation of knock out alleles by random Tol2 transposon insertional mutagenesis with a floxed/FRT gene trap, which can be reverted by Cre or Flip-mediated recombination (*Clark et al., 2011*; *Ni et al., 2012*; *Trinh et al., 2011*). Only recently have gene editing methods been developed in zebrafish to simplify Cre driver isolation (*Almeida et al., 2021*; *Kesavan et al., 2018*) and generate genuine lox-regulated conditional alleles by targeted integration (*Burg et al., 2018*; *Han et al., 2021*; *Hoshijima et al., 2016*; *Li et al., 2019*; *Sugimoto et al., 2017*). These advances in zebrafish Cre/lox genetics are being driven by rapidly evolving methods for homology directed gene editing using CRISPR-Cas9 and TALEN endonucleases.

We and others recently demonstrated CRISPR-Cas9 targeted integration is an efficient method to isolate zebrafish proneural specific Cre and CreERT2 drivers that are expressed under the control of endogenous gene regulatory elements (*Almeida et al., 2021*; *Kesavan et al., 2018*). Efforts to introduce loxP sequences directly into the zebrafish genome by somatic gene targeting was first described using TALENs to target a double-strand break in the cdrh2 gene, followed by repair from a DNA oligonucleotide template containing homologous sequences flanking a loxP site (*Bedell et al., 2012*). This approach was expanded upon using CRISPR-Cas9 to create a floxed allele in two zebrafish genes by sequential loxP oligonucleotide targeting at sites flanking an exon, and demonstrated robust gene knockdown and loss of function after Cre-mediated exon excision (*Burg et al., 2018*). Although this approach is effective for creating a Cre/lox regulated allele, a single integration event of a floxed cassette with a linked reporter has distinct advantages, including only one generation of targeting and fluorescent genotyping of the integration allele. The first example of a germline floxed conditional allele with linked reporter used a single TALEN site downstream of an exon (*Hoshijima et al., 2016*). The repair template consisted of a linear cassette containing 1 kb long homology arms flanking a floxed exon plus reporter, to replace the endogenous exon by homologous recombination. A similar strategy using long homology arms was used to isolate a zebrafish conditional shha allele by integration of a modified Flex invertible gene trap (*Ni et al., 2012*), called Zwitch, by homologous recombination from an intact plasmid (*Sugimoto et al., 2017*). Stable inversion of the Zwitch cassette by Cre led to effective shha gene knockdown and defective shha signaling. More recently homology-independent targeting strategies have been described to generate conditional knockouts with a linked reporter, or bidirectional knockin of a dual reporter gene trap cassette, providing a method for labeling conditional gene knockout cells (*Han et al., 2021*; *Li et al., 2019*). As an alternative to isolation of stable germline conditional alleles, rapid tissue-specific gene knockdown and cell labeling can be achieved by somatic targeting with a floxed Cas9-2A-GFP; U6:gRNA transposon in a Cre-specific transgenic background, allowing for simultaneous bi-allelic inactivation with GFP cell labeling (*Hans et al., 2021*). Together, these approaches provide effective conditional gene knockdown strategies with varying degrees of complexity in design and execution. However, there is still a need for a simple method to efficiently recover robust conditional alleles through a single gene engineering event.

To address this need for a simple method to recover zebrafish germline conditional alleles in a single generation, we applied our GeneWeld CRISPR-Cas9 knock in strategy (*Welker et al., 2021*;

*Wierson et al., 2020*) to target integration of a highly mutagenic floxed gene trap into an intron in two genes. We previously showed our GeneWeld strategy simplifies targeting vector assembly and enhances the efficiency of on target integration (*Almeida et al., 2021*; *Wierson et al., 2020*). In the present study, we built a universal vector, UFlip, which is derived from the GeneWeld pGTag and pPRISM targeted integration vectors (*Almeida et al., 2021*; *Wierson et al., 2020*), and the Tol2 Gene Breaking Transposon vectors (*Clark et al., 2011*; *Ichino et al., 2020*). The GeneWeld vectors contain sites for cloning short homology arms flanking a cargo of interest, and universal gRNA sites for efficient CRISPR-Cas9 cutting in vivo. In the UFlip vector, homology arm cloning sites flank a floxed gene trap cassette that contains a 2A-mRFP primary reporter based on the Tol2 RP2 gene trap (*Clark et al., 2011*; *Ichino et al., 2020*). The cassette also contains a heart or lens BFP secondary reporter for transgenic identification. Using 48 bp flanking short homology arms to drive integration at unique intron target gRNA sites, we recovered precise UFlip integration alleles in *rbbp4* and *rb1* with frequencies of 4–14%. We demonstrate that each molecular component in the floxed UFlip cassette functions as expected in vivo. Integration alleles with UFlip in the active, gene 'off' orientation show mRFP expression from the gene trap in the expected gene-specific pattern. UFlip 'off' alleles lead to >99% knock down of gene expression compared to wild-type levels and recapitulate indel mutant phenotypes, demonstrating highly effective loss of function by primary transcript splicing into the gene trap. Recombination of an 'off' allele by Cre injection or a transgenic Cre source leads loss of RFP expression, stable inversion of the UFlip cassette to the passive orientation, and robust conditional rescue of mutant phenotypes. UFlip 'on' alleles are phenotypically normal in combination with loss-of-function mutations. Recombination of an 'on' allele by Cre injection or a transgenic Cre source leads to induction of RFP expression, stable cassette inversion, and robust conditional inactivation. These results demonstrate GeneWeld UFlip CRISPR targeted integration is an effective approach to generate zebrafish Cre/*lox* conditional alleles that can be used for cell type-specific genetic mosaic analysis. The ability to perform conditional inactivation as well as conditional rescue provides a significant advance in zebrafish genetics.

## Results

### UFlip, a universal vector to generate stable Cre/*lox* regulated conditional alleles by CRISPR-Cas9 targeted integration

The Universal Flip (UFlip) vector was designed to be used with the GeneWeld strategy for CRISPR-Cas9 targeted integration driven by short homology (*Wierson et al., 2020*). The vector contains a cassette with a floxed gene trap plus secondary marker flanked by cloning sites for homology arms (HA) complementary to the genomic CRISPR target site, and universal gRNA sites (UgRNA) for in vivo liberation of the cassette after injection into zebrafish embryos (*Figure 1A*). The gene trap was derived from the gene-breaking Tol2 transposon RP2 that has previously been shown to lead to >99% knockdown of gene expression (*Clark et al., 2011*; *Ichino et al., 2020*). RP2 was modified by insertion of the porcine teschvirus-1 2A peptide sequence in between the splice acceptor and mRFP, which enables expression of a primary fluorescent reporter from the gene trap without creating a fusion protein. The ocean pout antifreeze gene transcriptional terminator follows the mRFP cDNA. Like our recently published endogenous Cre driver lines (*Almeida et al., 2021*), the primary gene trap reporter is followed by a secondary marker that drives tissue-specific expression of blue fluorescent protein (BFP) from the *Xenopus* gamma crystallin 1 (*gcry1*) lens promoter or the zebrafish myosin light chain 7 (*myl7*) cardiac muscle promoter, simplifying identification of an integration allele. Alternating pairs of *loxP* and *lox2272* sites oriented head to head flanking the cassette are designed to drive two Cre recombination events that invert and lock the cassette in place, a design strategy described previously to generate floxed conditional alleles in mice (*Robles-Oteiza et al., 2015*; *Schnütgen et al., 2003*) and zebrafish (*Sugimoto et al., 2017*). *rox* sites flanking the entire cassette provide an alternative method for inversion using Dre recombinase (*Figure 1A*), which we demonstrate below provides highly efficient recovery of inverted alleles through the germline. Integration of UFlip into an intron in the active, gene 'off' orientation is expected to lead to premature transcriptional termination of the primary transcript in the gene trap and splicing of upstream exons into 2A-mRFP (*Figure 1B*), resulting in a loss-of-function allele. In contrast, integration of UFlip into an intron in the passive, gene 'on' orientation is not predicted to interrupt endogenous gene expression, since RNA polymerase will

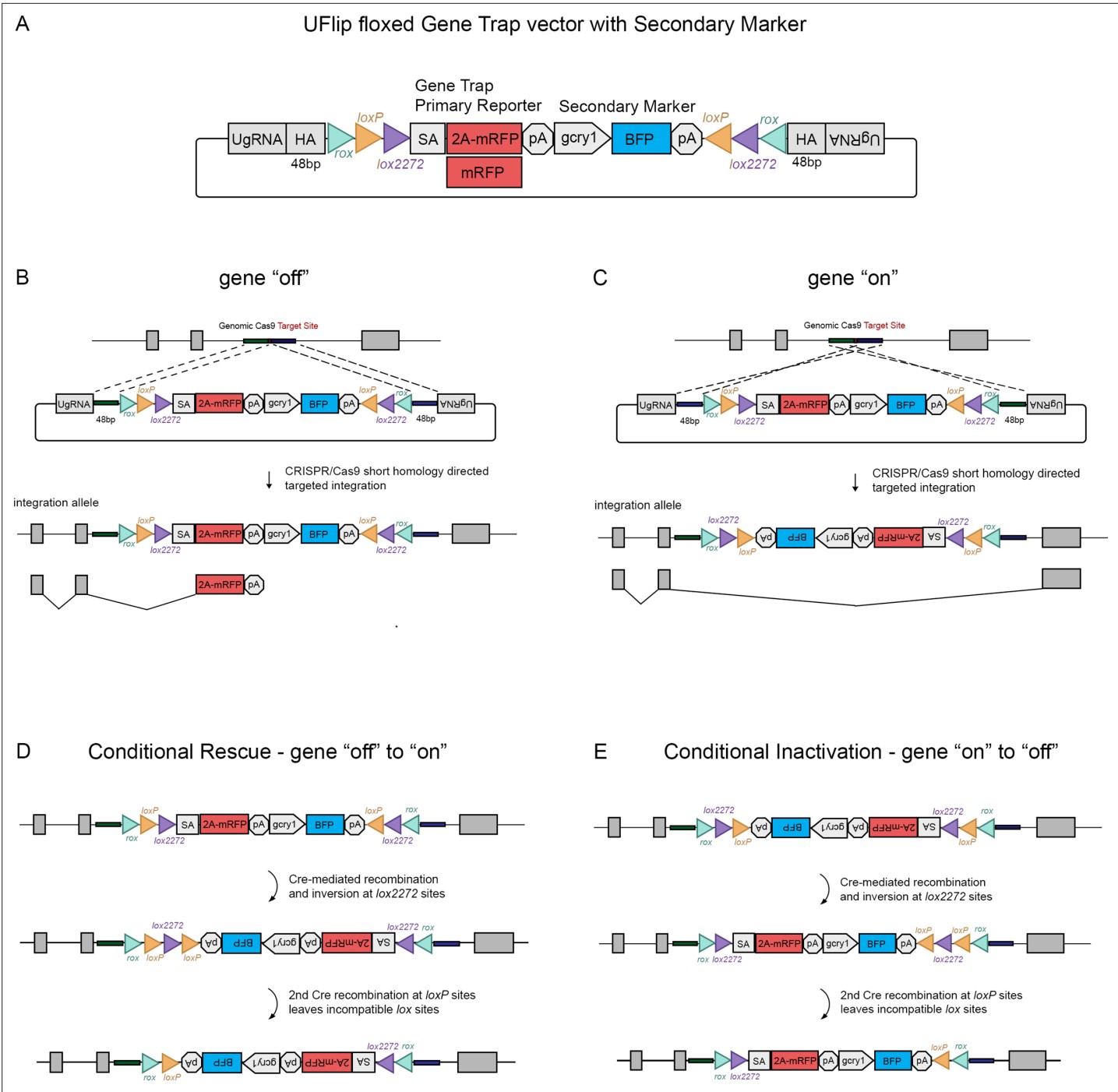

**Figure 1.** The UFlip floxed gene trap vector for isolation of conditional gene alleles generated by GeneWeld CRISPR-Cas9 targeted integration. (**A**) Diagram of the UFlip. The vector contains a floxed *rox loxP lox2272* gene trap plus secondary marker *loxP lox2272 rox* cassette. The cassette is flanked by cloning sites for homology arms (HA) complementary to a genomic CRISPR target site, and universal gRNA sites (UgRNA) for in vivo liberation of the targeting cassette. (**B**) Gene 'off' alleles are generated by integration of the UFlip cassette into an intron in the active orientation, leading to transcription termination and splicing of the primary transcript in the mRFP gene trap. (**C**) Gene 'on'' alleles are generated by integration of the UFlip cassette into an intron in the passive orientation. This is driven by cloning the genomic 5' homology arm downstream of the UFlip cassette, and cloning the genomic 3' homology arm upstream of the UFlip cassette. Integration at the genomic CRISPR-Cas9 target site occurs in the opposite orientation. During transcription RNA polymerase reads through the integrated UFlip cassette, which is spliced out with the intron during processing of the primary transcript. (**D**) Cre-mediated recombination at an 'off' allele locks the cassette in the 'on' orientation. The first recombination occurs stochastically at either *lox2272* or *loxP* sites. The diagram shows the intermediate that forms if the first recombination occurs at the *lox2272* sites. (**E**) Cre-mediated recombination at an 'on' allele locks the cassette in the 'off' orientation. The first recombination occurs stochastically at either *lox2272* or *loxP*

*Figure 1 continued on next page*

*Figure 1 continued*

sites. The diagram shows the intermediate that forms if the first recombination occurs at the *lox2272* sites. BFP, blue fluorescent protein; *gcry1*, gamma crystallin 1 promoter; *myl7*, cardiac myosin light chain 7 promoter; 2 A, porcine teschvirus-1 2A peptide; mRFP, monomeric red fluorescent protein; pA, transcription termination and polyadenylation signal; SA, splice acceptor.

read through the cassette in the intron, which is then spliced out of the mature transcript (*Figure 1C*). For clarity, the UFlip alleles described here are referred to as 'off' when the cassette was integrated in the active, gene trap orientation. Alleles are designated 'on' when UFlip was integrated in the passive orientation, that leads to removal of the UFlip cassette during splicing and allows proper functioning of the gene.

The process by which Cre recombination leads to stable inversion of the UFlip cassette is illustrated in *Figure 1D, E* (*Schnütgen et al., 2003*). Introduction of Cre leads to stochastic recombination at either the *loxP* or the *lox2272* pairs of recombinase sites. *Figure 1D, E* show the intermediate results after 1 recombination event at the *lox2272* sites. In the intermediate the cassette has inverted from 'off' to 'on' (*Figure 1D*) or from 'on' to 'off' (*Figure 1E*), resulting in a pair of *loxP* sites flanking one of the *lox2272* sites. Recombination at the head to tail oriented *lox*P sites removes the intervening *lox2272*, resulting in a cassette flanked by a single pair of incompatible *loxP* and *lox2272* sites. This prevents further recombination and produces stable inversion of the cassette.

## Isolation of *rbbp4* and *rb1* UFlip conditional alleles

To test the ability of the UFlip cassette to generate zebrafish Cre-responsive conditional alleles, we chose to target integration of UFlip two genes, *retinoblastoma binding protein 4* (*rbbp4*) and *retinoblastoma 1* (*rb1*). We had previously isolated indel mutations in both genes that show strong loss-of-function phenotypes at the morphological and cellular level (*Schultz et al., 2018*; *Schultz-Rogers et al., 2022*; *Solin et al., 2015*). Intronic genomic DNA was amplified from wild-type WIK fish, cloned, sequenced, and analyzed to locate unique Cas9 gRNA sites that were shared among fish and that did not map to repetitive elements or retrotransposons. Cas9 target sites were identified in *rbbp4* intron 4 and *rb1* intron 6 (*Table 1*) and synthetic guides ordered from Synthego (https://www.synthego.com). Efficient indel formation at the intronic gRNA sites was confirmed by co-injection of gRNA and Cas9 mRNA into single-cell embryos, followed by extraction of genomic DNA at 2 dpf. A PCR amplicon surrounding the target site was directly sequenced followed by ICE analysis (https://ice.synthego.com/#/). The *rbbp4* intron 4 and *rb1* intron 6 guides showed 50% and 95% indel formation at the target site, respectively (*Figure 2A, B*). UFlip targeting vectors were assembled with 48 bp 5' and 3' homology arms complementary to the DNA flanking the Cas9 genomic DNA double strand break site (*Table 1*). One cell stage WIK embryos were co-injected with targeting vector (10 pg), Cas9 mRNA (150 pg), universal gRNA (25 pg), and gene-specific gRNA (25 pg). Four to six embryos with positive expression of the primary mRFP reporter and secondary BFP reporter were selected to test for evidence of on target integration by PCR amplification of the 5' and 3' junctions (data not shown). Primary and secondary reporter positive siblings were raised to adulthood. *rbbp4* intron 4 was targeted with the UFlip-2A-mRFP cassette in the active orientation to recover an 'off' allele. Embryos expressing the primary reporter mRFP in the nervous system, where *rbbp4* is normally expressed, and the lens:BFP secondary marker, were raised to adulthood. Two out of seven adult F0 fish transmitted embryos with widespread mRFP and lens BFP expression (*Table 2*; *Table 2—source data 1*). One adult transmitted an allele with precise 5' and 3' junctions at the intron target site. F1 embryos were raised to adulthood and the presence of the precise *rbbp4-2A-mRFP-off, gcry1:BFP* (*rbbp4*$^{off}$) allele

**Table 1.** Genome intronic CRISPR gRNA sites and UFlip targeting vector homology arm sequences.

| Gene | Genomic sgRNA with PAM | 5' Homology arm | 3' Homology arm |
|---|---|---|---|
| *rbbp4* | GTTGAAGGATTAAGAGTTAAGGG | AAAATCTGCTAGCTGTATATTGTTCTTATTTGAT GAAGAAGACCCTTA | ACTCTTAATCCTTCAACTTCGTTGCAAAAAAGTCA GTTGTGTAAAGGT |
| *rb1* | ATTAGAAGAGAGTCCCAATGGGG | CCTTGATTACAGTTTCTGCTTTTGTGAG TGTACTGTAGTTTGCCCTAA | TCTTCTCTCAGGGTTACATGTTTAATGGATAGTGTGT CCATGTTGTCA |

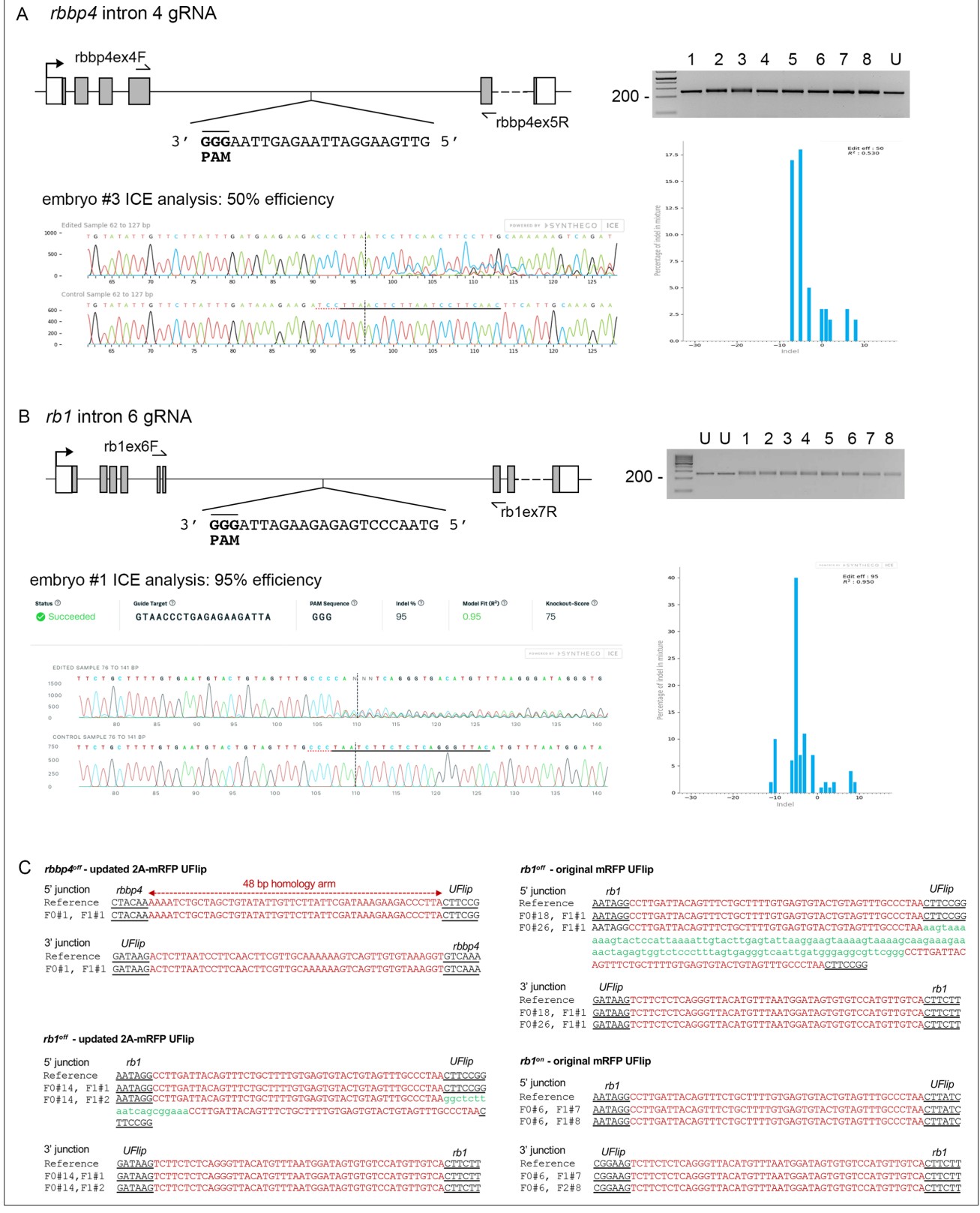

**Figure 2.** *rbbp4* and *rb1* intronic gRNA efficiency and F1 UFlip allele junction analysis. (**A**) *rbbp4* gene model with sequence of the intron 4 reverse strand gRNA. Gel image of PCR amplicons surrounding the target site from 8 Cas9 plus gRNA injected and 1 uninjected (U) embryo. Amplicons from embryo #3 and the uninjected embryo were sequenced and analyzed with Synthego's ICE software, and indicate 50% indel efficiency at the target site. Plot shows the range and percentage of indels present in the sequences. PAM sequence shown in bold and underlined. (**B**) *rb1* gene model with

*Figure 2 continued on next page*

*Figure 2 continued*

sequence of the intron 6 reverse strand gRNA. Gel image of PCR amplicons surrounding the target site from eight embryos injected with Cas9 and the gRNA (1-8), and two uninjected embryos (U). Amplicons from embryo #1 and an uninjected embryo were sequenced and analyzed with Synthego's ICE software, and indicate 95% indel efficiency at the target site. Plots show the range and percentage of indels present in the sequences. PAM sequences shown in bold and underlined. (**C**) 5′ and 3′ genomic-UFlip integration junctions were PCR amplified from F1 transgenic zebrafish fin clip genomic DNA. The PCR products were sequenced and aligned to the reference sequence expected for a precise integration at the genomic target site. Capitalized red nucleotides represent 48 bp homology arms. Lowercase green nucleotides represent random inserted sequences.

confirmed by fin clip and sequencing (*Figure 2C*). An F1 adult with the precise integration allele was outcrossed to WIK to establish an F2 family of the *rbbp4*$^{off\ is61}$ allele.

We initially targeted *rb1* intron 6 with the original UFlip-mRFP primary reporter construct designed to create an in-frame fusion of the polypeptide with mRFP. We recovered *rb1-mRFP-off, gcry1:BFP (rb1*$^{off}$*)* and *rb1-mRFP-on, gcry1:BFP (rb1*$^{on}$*)* alleles with frequencies of 6% (2/33) for *rb1*$^{off}$ and 4% (1/25) for *rb1*$^{on}$ (*Table 2*; *Table 2—source data 1*). 5′ and 3′ junction PCR and sequence analysis of F1 embryos showed the integrations were precise, except for the 5′ junction in the allele transmitted by the second *rb1*$^{off}$ founder #26 (*Figure 2C*). The 5′ junction included a duplication of the 5′ Homology Arm sequence that flanked a 115 bp segment of the vector backbone. The *rb1*$^{off}$ founder #18 transmitting the precise allele was lost, therefore we moved forward with the allele from founder #26. As shown below, the 153 bp insertion on the 5′ side of the *rb1*$^{off}$ integration did not appear to affect the functionality of the allele. F1 adults were outcrossed to wildtype WIK to establish *rb1*$^{off\ is58}$ and *rb1*$^{on\ is57}$ families. We subsequently targeted the same *rb1* intron six site with the updated UFlip-2A-mRFP vector, which allows expression of mRFP without creating a fusion protein with the targeted gene polypeptide. We recovered 2/8 (12.5%) founders transmitting lens:BFP +alleles with ubiquitous expression of the mRFP primary reporter (*Table 2*; *Table 2—source data 1*). Founder F0#14 transmitted two alleles, one with precise junctions, the other with a precise 3′ junction and a 5′ junction with a homology arm duplication and insertion of 19 bp (*Figure 2C*). F1s from this founder will be used to establish the line *rb1*$^{off\ is63}$. For the experiments outlined below demonstrating Cre-mediated *rb1* conditional rescue and inactivation, we used the established F3 generation *rb1*$^{off\ is58}$ and *rb1*$^{on\ is57}$ lines generated with the original UFlip construct, which did not show the mRFP gene trap reporter expression.

The UFlip construct contains *rox* sites flanking the entire cassette to allow the use of Dre recombination to recover an allele of the opposite orientation when starting with the *rbbp4*$^{off}$ allele (*Figure 3A*). To test this, we injected *Dre* synthetic mRNA into F1 and F2 *rbbp4*$^{off/+}$ embryos. Injected embryos were raised to adulthood and five F1 and six F2 adults were outcrossed to wildtype WIK. Embryos were screened for expression of the primary mRFP reporter and the linked BFP secondary marker. Three of 5 F1 adults, and 4/6 F2 adults, transmitted both the mRFP/BFP positive original 'off' allele to their progeny, as well as the mRFP negative/BFP positive Dre-inverted 'on' allele (*Table 3*). The frequency of embryos inheriting the inverted allele vs. the original allele ranged from 7.7% to 43.2% (*Table 3*). We confirmed the *rbbp4*$^{on}$ allele recovered by Dre recombination contained the expected 5′ and 3′ junctions by PCR amplification and sequencing (*Figure 3B, C*). A single F2 adult was used to establish an F3 family of the *rbbp4*$^{on\ is62}$ allele. These results demonstrate efficient Dre mediated recombination at *rox* sites in the germline, and a simple method for isolation of an alternative conditional allele starting with one UFlip allele.

**Table 2.** Recovery of *rbbp4* and *rb1* UFlip floxed conditional alleles by GeneWeld CRISPR-Cas9 targeted integration.

| UFlip allele | Targeted Intron | Homology arm length bp/bp | Injected F0 embryo secondary marker expression | Germline transmission of secondary marker | Germline transmission of precise integration allele |
|---|---|---|---|---|---|
| *rbbp4*$^{off\ is61}$ | 4 | 48/48 | 19% (26/134) | 29% (2/7) | 14% (1/7) |
| *rb1*$^{off\ is58}$ | 6 | 48/48 | 47% (34/73) | 9% (3/33) | 6% (2/33) |
| *rb1*$^{off\ is63}$ | 6 | 48/48 | 48% (60/125) | 25% (2/8) | 12.5% (1/8) |
| *rb1*$^{on\ is57}$ | 6 | 48/48 | 43% (61/101) | 20% (5/25) | 4% (1/25) |

The online version of this article includes the following source data for table 2:

**Source data 1.** *rbbp4* and *rb1* UFlip embryo injection secondary marker F0 screening data and germline transmission data.

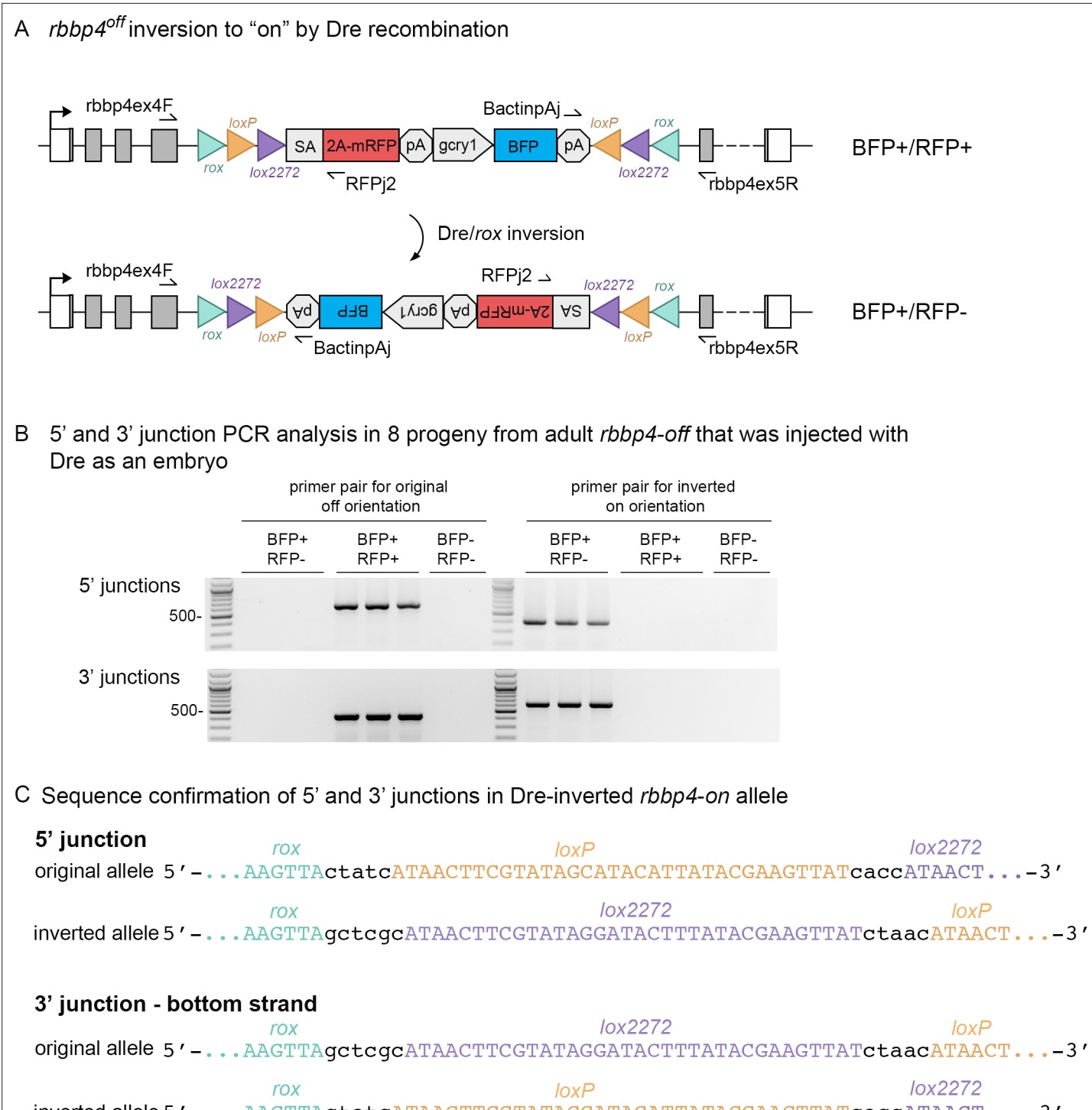

**Figure 3.** *Dre* mRNA injection into *rbbp4^off/+* embryos leads to inversion of the UFlip cassette and efficient germline recovery of an inverted *rbbp4^on* allele. (**A**) Diagram illustrating Dre-mediated inversion of the *rbbp4^off* allele to the on orientation. Repeated inversion of the cassette will continue as long as Dre is present. The final allele is predicted to be in the inverted 'on' orientation at a frequency of 50%. (**B**) PCR junction analysis of 8 embryos from an F1 adult that had been injected with *Dre* mRNA at the one-cell stage. Three embryos positive for expression of the lens BFP secondary marker show the expected 5' and 3' junction PCR amplicons for the inverted *rbbp4^on* allele. (**C**) Sequence analysis confirms Dre-mediated inversion of the cassette from the 'off' to 'on' orientation in BRP+/RFP - embryos.

**Table 3.** Germline recovery of *rbbp4*<sup>on</sup> allele by Dre-mediated inversion of *rbbp4*<sup>off</sup>.

| Adult *rbbp4*<sup>off/+</sup> | BFP-/RFP-embryos | BFP+/RFP + embryos (Off) original allele | BFP+/RFP- embryos (On) inverted allele | % transmission of On inverted allele/ total BFP+ |
|---|---|---|---|---|
| F1 Female #1 | 48 | 37 | 0 | 0 |
| F1 Female #2 | 76 | 62 | 11 | 15% (11/73) |
| F1 Female #3 | 25 | 16 | 5 | 24% (5/21) |
| F1 Female #4 | 20 | 13 | 2 | 13% (2/15) |
| F1 Female #5 | 19 | 12 | 0 | 0 |
| F2 Female #1 | 18 | 24 | 2 | 7.7% (2/26) |
| F2 Female #2 | 50 | 25 | 19 | 43.2% (19/44) |
| F2 Female #3 | 36 | 40 | 4 | 9.1% (4/44) |
| F2 Female #4 | 22 | 14 | 0 | 0 |
| F2 Male #1 | 139 | 87 | 14 | 13.9% (14/101) |
| F2 Male #2 | 37 | 28 | 0 | 0 |

Experimental group 1: F1 embryos from the F0 founder *rbbp4*<sup>off/+</sup> adult crossed to WIK were injected with 15 pg *Dre* mRNA. Inversion was confirmed by PCR on genomic DNA isolated from -injected embryos . 5 sibling injected embryos were raised to adulthood and were outcrossed to WIK. F2 embryos were screened for Primary (RFP+) and Secondary (BFP+) marker expression. Embryos inheriting the original *rbbp4*<sup>off</sup> allele show expression of both markers. Embryos inheriting an allele that was inverted from "off" to "on" by Dre recombination show expression of the secondary marker (BFP+) but lose expression of the primary marker due to the inverted orientation of the cassette. **Experimental group 2**: The same experiment was performed with an adult F1 *rbbp4*<sup>off/+</sup> female crossed to WIK. F2 *Dre* mRNA injected embryos were raised to adulthood, and 7 adults screened for transmission of the inverted *rbbp4*<sup>on</sup> allele to the F3 generation.

## *rbbp4*<sup>off</sup> allele provides robust gene knockdown and loss of function

To demonstrate that integration of the UFlip cassette in the active, 'off' orientation leads to effective gene knockdown we examined gene expression by quantitative PCR in *rbbp4*<sup>off/off</sup> homozygotes. Homozygous *rbbp4*<sup>off/off</sup> 3 dpf larvae show a>99% reduction in *rbbp4* mRNA levels in comparison to wild type sibling larvae (*Figure 4A, B*; *Figure 4—source data 1*). In comparison to wild-type larvae, the *rbbp4*<sup>off</sup> allele is easily identifiable by primary reporter gene trap RFP expression in the retina, brain, and mesodermal tissues, and secondary marker lens specific BFP expression (*Figure 4C, D* arrowhead). In trans-heterozygous combination with the established loss-of-function indel allele *rbbp4*<sup>Δ4</sup> (*Figure 4D, E*), the *rbbp4*<sup>off</sup> allele recapitulates the microcephaly and microphthalmia phenotypes found in *rbbp4*<sup>Δ4/Δ4</sup> homozygotes (*Schultz et al., 2018*; *Schultz-Rogers et al., 2022*). In comparison to control, anti-activated caspase 3 labeling and quantification in sectioned *rbbp4*<sup>off/Δ4</sup> 2 dpf head tissue reveals extensive apoptosis throughout the midbrain (** $p < 0.01$) and retina (** $p < 0.01$) (*Figure 4F, G*). This cellular phenotype is not significantly different from our previous description of apoptosis in *rbbp4*<sup>Δ4/Δ4</sup> homozygotes (*Figure 4F–H*). Together these results demonstrate the *rbbp4*<sup>off</sup> allele causes robust gene knockdown by the primary reporter RFP gene trap, induces the expected pattern of RFP expression, and recapitulates the *rbbp4* strong loss of phenotype at the molecular, cellular, and morphological levels.

## *rbbp4*<sup>on</sup> allele maintains *rbbp4* function

In the *rbbp4*<sup>on</sup> allele, the UFlip cassette is predicted to be read through by RNA polymerase and then removed from the primary transcript when intron 4 is spliced out (*Figure 4I*). To test whether the *rbbp4*<sup>on</sup> allele alters normal *rbbp4* mRNA expression levels RT-qPCR was performed on 3 dpf *rbbp4*<sup>on/+</sup> heterozygous and *rbbp4*<sup>on/on</sup> homozygous larvae. *rbbp4*<sup>on/+</sup> heterozygotes showed ~25% reduction in wild-type *rbbp4* mRNA levels, while *rbbp4*<sup>on/on</sup> homozygotes showed ~40% reduction (*Figure 4J*, *Figure 4—source data 1*). This suggests the integrated UFlip cassette may impact splicing rate, stability, or other processing of the primary transcript. However, neither *rbbp4*<sup>on/+</sup> heterozygotes nor *rbbp4*<sup>on/Δ4</sup> trans-heterozygous show a gross phenotype in 5 dpf larvae (*Figure 4K and L*) or evidence of apoptosis in the developing midbrain and retina at 2 dpf (*Figure 4M and N*). Therefore, the level of processed, mature *rbbp4* transcript is enough to maintain wild type levels of *rbbp4* activity.

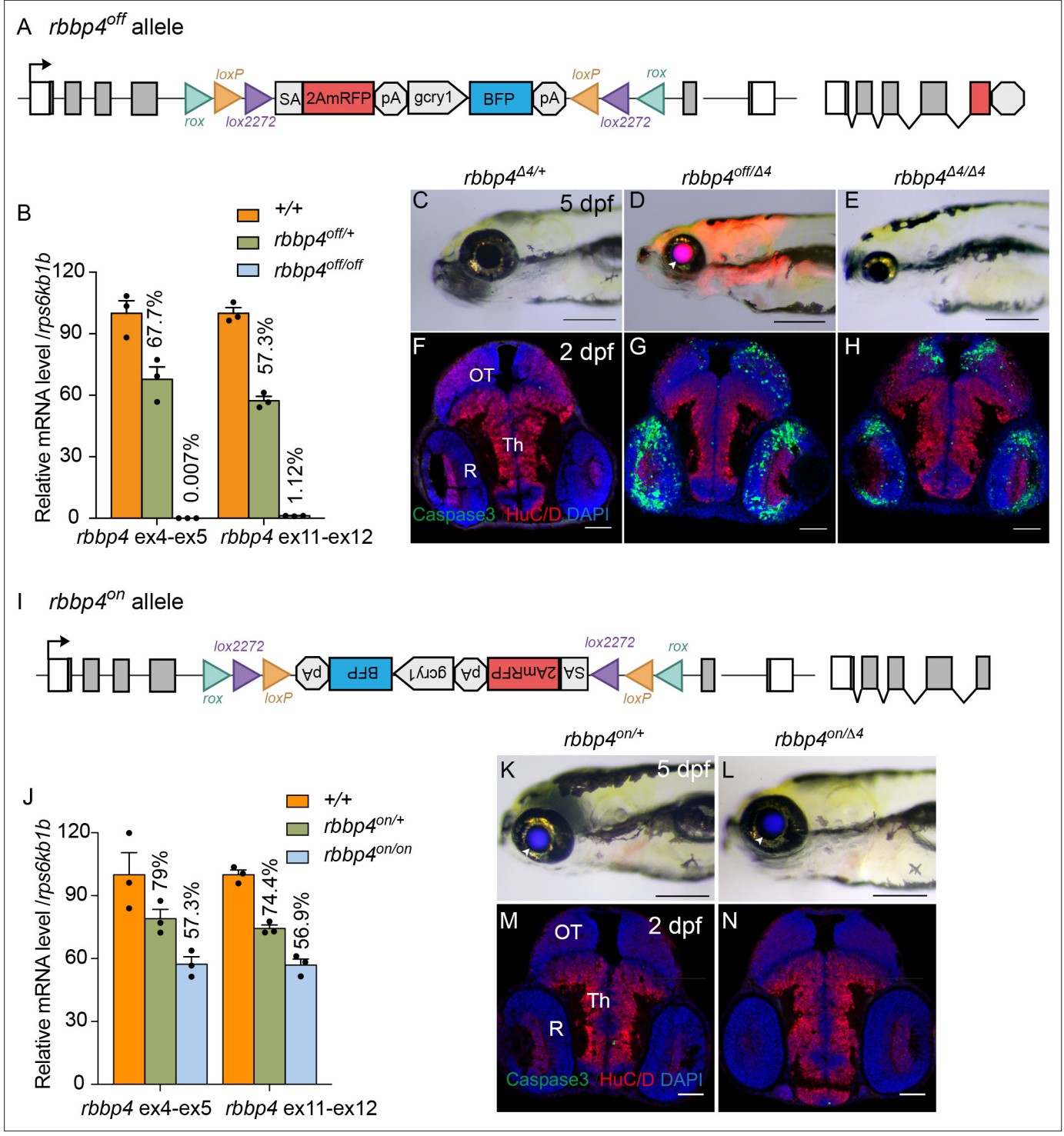

**Figure 4.** Molecular and phenotypic characterization of *rbbp4^off^* and *rbbp4^on^* alleles. (**A**) Diagram of the *rbbp4^off^* allele. (**B**) Plot of RT-qPCR results from wild type +/+ (n=3), heterozygous *rbbp4^off/+^*(n=3), and homozygous *rbbp4^off/off^* (n=3) larvae showing the relative level of *rbbp4* mRNA transcript using reference gene *rps6kb1b*. Primer pairs were located in exons 4 and 5, or downstream exons 11 and 12. (**C – E**) Gross phenotype of *rbbp4^Δ4/+^* (**C**), *rbbp4^off/Δ4^* (**D**), and *rbbp4^Δ4/Δ4^* (**E**) 5 dpf larvae. Arrowhead in (**D**) points to overlap of *rbbp4^off^ 2A-mRFP* primary reporter and *gcry1:BFP* secondary reporter expression in the lens, which appears purple. (**F – H**) Caspase-3 and HuC/D labeling of sectioned head tissue from 2 dpf *rbbp4^Δ4/+^* (**F**) *rbbp4^off/Δ4^* (**G**) and *rbbp4^Δ4/Δ4^* (**H**) embryos. (**I**) Diagram of the *rbbp4^on^* allele. (**J**) Plot of RT-qPCR results from wild type +/+ (n=3), heterozygous *rbbp4^on/+^* (n=3), and homozygous *rbbp4^on/on^* (n=3) larvae showing the relative level of *rbbp4* mRNA transcript using reference gene *rps6kb1b*. Primer pairs were located in exons 4 and 5, or downstream exons 11 and 12. (**K, L**) Gross phenotype of *rbbp4^on/+^* (**K**) and *rbbp4^on/Δ4^* (**L**) 5 dpf larvae. The *rbbp4^on^* allele secondary

*Figure 4 continued on next page*

Figure 4 continued

marker *gcry1:BFP* expression is visible in the lens. Caspase-3 and HuC/D labeling of sectioned head tissue from 2 dpf *rbbp4*$^{\Delta 4/+}$ (**M**) and *rbbp4*$^{off/\Delta 4}$ (**N**) embryos. OT, optic tectum; R, retina; Th, thalamic region. Error bars represent mean ± s.e.m. Scale bars: 200 μm (**C–E, K, L**), 50 μm (**F–H, M,N**).

The online version of this article includes the following source data and figure supplement(s) for figure 4:

**Source data 1.** Source data for *rbbp4* RT-quantitative PCR analysis in wildtype, heterozygous and homozygous embryos from *rbbp4*$^{off/+}$ and *rbbp4*$^{on/+}$ incrosses.

**Figure supplement 1.** Quantification of activated caspase-3a labeled cells in *rbbp4*$^{\Delta 4/+}$, *rbbp4*$^{off/\Delta 4}$, and *rbbp4*$^{\Delta 4/\Delta}$ 2 dpf embryo midbrain and retina.

**Figure supplement 1—source data 1.** Source data for quantification of activated caspase-3a labeled cells in *rbbp4*$^{\Delta 4/+}$, *rbbp4*$^{off/\Delta 4}$, and *rbbp4*$^{\Delta 4/\Delta}$ 2 dpf embryo midbrain and retina.

## Cre/*lox* recombination of *rbbp4*$^{off}$ leads to stable inversion and conditional gene rescue

To test the ability of the UFlip cassette to undergo stable inversion and provide conditional rescue, we first used *Cre* mRNA injection. A total of 12.5 pg *Cre* synthetic mRNA was injected into embryos from a cross between *rbbp4*$^{off/+}$ and *rbbp4*$^{\Delta 4/+}$ heterozygous adults (**Figure 5A–G**). In comparison to uninjected *rbbp4*$^{off/\Delta 4}$ larvae (**Figure 5B**), 5 dpf *rbbp4*$^{off/\Delta 4}$ *Cre* injected larvae show normal morphology and an absence of RFP expression (**Figure 5C**). Confocal live imaging of 2 dpf in injected *rbbp4*$^{off/\Delta 4}$ confirmed loss of RFP and rescue of the cell death phenotype in the retina (**Figure 5—figure supplement 1A–L**). Stable inversion of the *rbbp4*$^{off}$ cassette to the 'on' orientation was confirmed by PCR and sequence analysis, which showed the expected sequences at the 5' and 3' junctions after inversion at the *lox* sites (**Figure 5—figure supplement 1M–P**). Activated caspase 3 labeling in 2 dpf sectioned head tissue showed a nearly complete absence of apoptosis in the midbrain (** p<0.01) and retina (**** p<0.0001) compared to controls (**Figure 5D–F**, **Figure 5—source data 1**). Genomic DNA quantitative PCR of the original 5' and 3' *rbbp4*$^{on}$ allele junction fragments in the Cre injected embryos indicated a 93% reduction (**Figure 5G**, **Figure 5—source data 1**). Together these data demonstrate Cre mediated recombination at the incompatible pairs of *loxP* and *lox2272* sites in the *rbbp4*$^{off}$ allele leads to efficient, stable inversion of the cassette to the 'on' orientation, which results in robust conditional rescue of the *rbbp4* loss-of-function phenotype.

To test whether cell type-specific Cre sources could be used to rescue *rbbp4* activity in distinct cell populations (**Figure 5H–K**), we used our previously published endogenous proneural Cre driver lines *ascl1b-2A-Cre, gcry1:GFP*$^{is75}$ and *neurod1-2A-Cre, gcry1:GFP*$^{is77}$ (**Almeida et al., 2021**). 2 dpf larvae from *rbbp4*$^{off/+}$ crossed with *ascl1b-2A-Cre/+; rbbp4*$^{\Delta 4/+}$ were sectioned and head tissue labeled to detect activated caspase 3. In comparison to control *rbbp4*$^{off/\Delta 4}$ embryos, *ascl1b-2A-Cre; rbbp4*$^{off/\Delta 4}$ embryos appeared to have reduced caspase labeling in the midbrain, where *Cre* is expressed (**Figure 5H–I'**), but the change was not significant (n.s. p=0.3248) (**Figure 5K**; **Figure 5—source data 1**). *ascl1b* isn't expressed in the retina and consistent with this, Cre labeling was not detected in this tissue (**Figure 5I''**). As expected, there was no significant change in caspase labeling or levels in retinal sections from *ascl1b-2A-Cre; rbbp4*$^{off/\Delta 4}$ embryos (n.s. p=0.8152) (**Figure 5I'' and K**). To confirm expression of functional Cre recombinase and inversion of the cassette by *ascl1b-2A-Cre*, 5' and 3' junction genomic DNA qPCR showed >30% inversion of the cassette (**Figure 5—figure supplement 2**; **Figure 5—figure supplement 2—source data 1**). Testing for conditional rescue with the *neurod1-2A-Cre* driver showed there was no significant rescue of apoptosis in the brain (n.s. p=0.7794) and retina (n.s. p=0.9365) where *neurod1-2A-Cre* is expressed (**Figure 5J, K**; **Figure 5—source data 1**). 5' and 3' junction genomic DNA qPCR confirmed 30% inversion of the *rbbp4*$^{off}$ allele to the 'on' orientation (**Figure 5—figure supplement 2**; **Figure 5—figure supplement 2—source data 1**). Together, these results demonstrate a ubiquitous source of Cre can robustly rescue the loss-of-function phenotype of the *rbbp4*$^{off}$ allele. *ascl1b-2A-Cre* led to a modest rescue in the developing midbrain, possibly due to expression of *ascl1b* in a subset of proneural progenitors in this tissue. The complete absence of rescue by *neurod1-2A-Cre* indicates *rbbp4* isn't required for survival in this progenitor population. These interpretations are supported by the results of conditional gene inactivation experiments with the *rbbp4*$^{on}$ allele described below. *rbbp4*$^{on}$ cell-type-specific inactivation with *ascl1b-2A-Cre* leads to the expected apoptotic mutant phenotype in the midbrain, whereas inactivation with *neurod1-2A-Cre* did not result in apoptosis in either the midbrain or retina (**Figure 6**).

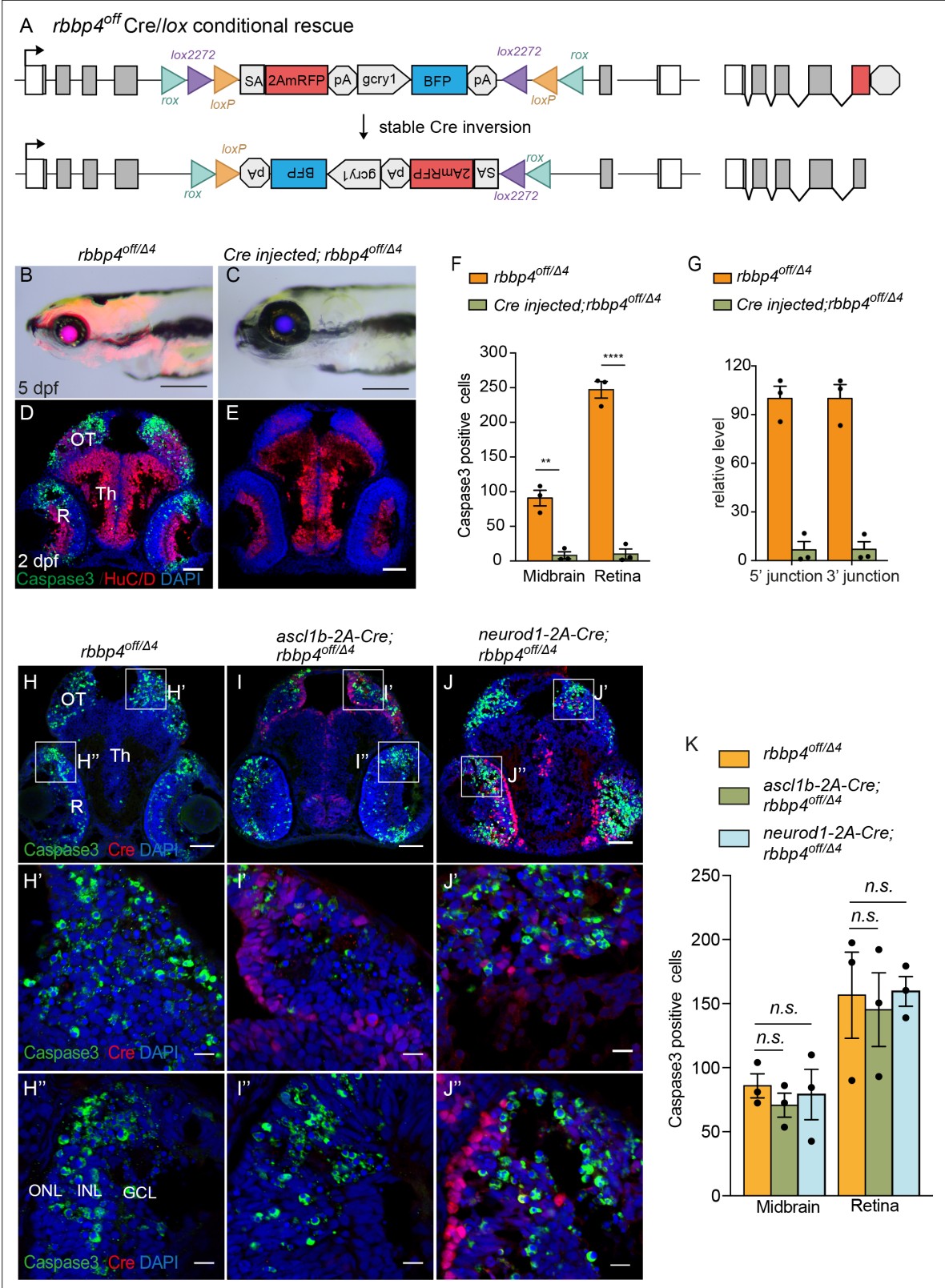

**Figure 5.** Ubiquitous and cell-type specific Cre-mediated conditional rescue with *rbbp4-off*. (**A**) Diagram of expected Cre mediated inversion of *rbbp4*<sup>off</sup> to on orientation. (**B**) Gross morphological phenotype of microcephaly and microphthalmia in 5 dpf transheterozygous *rbbp4*<sup>off/Δ4</sup>larva. (**C**) Cre injected 5 dpf transheterozygous *rbbp4*<sup>off/Δ4</sup>larva shows rescue of gross phenotype and loss of mRFP expression. (**D**) Activated caspase-3a labeling throughout midbrain and retina section from 2 dpf transheterozygous *rbbp4*<sup>off/Δ4</sup>embryo. (**E**) Absence of activated caspase-3a labeling in midbrain and retina of 2 dpf

*Figure 5 continued on next page*

Figure 5 continued

transheterozygous *rbbp4$^{off/\Delta4}$*embryo after Cre injection. (**F**) Quantification of caspase-3a labeling in control *rbbp4$^{off/\Delta4}$*(n=3) and Cre injected *rbbp4$^{off/\Delta4}$* (n=3) midbrain (** p<0.01) and retina (**** p<0.0001). (**G**) Genomic DNA qPCR quantification of *rbbp4$^{off}$* original orientation 5' and 3 junctions in control *rbbp4$^{off/\Delta4}$* (n=3) and Cre injected *rbbp4$^{off/\Delta4}$* (n=3). Cre injection reduced the level of *rbbp4$^{off}$* original orientation 5' (>93%) and 3' junctions (>93%). (**H –J"**) Activated caspase-3A and Cre labeling in sectioned head tissue from 2 dpf *rbbp4$^{off/\Delta4}$* (**H-H"**), *ascl1b-2A-Cre; rbbp4$^{off/\Delta4}$* (**I-I"**), and *neurod1-2A-Cre; rbbp4$^{off/\Delta4}$* (**J-J"**) embryos. (**K**) Quantification of caspase-3a labeling in *rbbp4$^{off/\Delta4}$*(n=3), *ascl1b-2A-Cre; rbbp4$^{off/\Delta4}$* (n=3) and *neurod1-2A-Cre; rbbp4$^{off/\Delta4}$* (n=3).*rbbp4$^{off/\Delta4}$*vs. *ascl1b-2A-Cre; rbbp4$^{off/\Delta4}$* midbrain (n.s. p=0.3248) and retina (n.s. p=0.8153), and *neurod1-2A-Cre; rbbp4$^{off/\Delta4}$* midbrain (n.s. p=0.7794) and retina (n.s. p=0.9365). OT, optic tectum; R, retina; Th, thalamic region. Error bars represent mean ± s.e.m. with two-tailed *t*-test. Scale bars: 200 μm (**B, C**), 50 μm (D, E, H – J). 10 μm (H' – J").

The online version of this article includes the following source data and figure supplement(s) for figure 5:

**Source data 1.** Source data for quantification of activated caspase-3a labeling and *rbbp4$^{off}$* inversion after Cre injection, activated caspase-3a labeling using *ascl1b-2A-Cre* and *neurod1-2A-Cre*.

**Figure supplement 1.** Live imaging and molecular analysis of *rbbp4$^{off}$* conditional rescue by Cre injection.

**Figure supplement 2.** qPCR quantification of *rbbp4$^{off}$* allele inversion by *ascl1b-2A-Cre* and *neurod1-2A-Cre*.

**Figure supplement 2—source data 1.** Source data for qPCR quantification of *rbbp4$^{off}$* inversion by *ascl1b-2A-Cre* and *neurod1-2A-Cre*.

## Cre/*lox* recombination of *rbbp4$^{on}$* leads to efficient conditional gene inactivation and recapitulates morphological and cellular loss-of-function phenotypes

To assess whether Cre recombination of the *rbbp4$^{on}$* allele to the 'off' orientation would lead to conditional inactivation and induction of expression of the gene trap mRFP, we first used 12.5 pg *Cre* mRNA injection into embryos from a cross between *rbbp4$^{on/+}$* and *rbbp4$^{\Delta4/+}$* heterozygous adults (**Figure 6**). In comparison to uninjected *rbbp4$^{on/\Delta4}$* larvae which appear morphologically normal (**Figure 6B**), 5 dpf *rbbp4$^{on/\Delta4}$* *Cre* mRNA-injected larvae display the *rbbp4* microcephaly and microphthalmia loss-of-function phenotype, and induction of RFP expression (**Figure 6C**). Sectioned tissue from 2 dpf embryos showed a significant increase in activated caspase 3 labeling throughout the midbrain (* p<0.05) and retina (* p<0.05) (**Figure 6D–F**; **Figure 6—source data 1**), consistent with the *rbbp4* mutant apoptotic cellular phenotype. Confocal live imaging in 2 dpf control and injected *rbbp4$^{on/\Delta4}$* embryos confirmed induction of the gene trap RFP expression and cell death in the retina (**Figure 6—figure supplement 1A–L**). PCR junction analysis and sequencing (**Figure 6—figure supplement 1M–P**), and genomic DNA qPCR analysis of 5' and 3' junctions (**Figure 6G**; **Figure 6—source data 1**) confirmed robust stable inversion of the *rbbp4$^{on}$* allele to the 'off' orientation (82%–84% efficiency) correlated with induction of the *rbbp4* loss-of-function phenotype.

## Proneural Cre/*lox rbbp4$^{on}$* conditional inactivation reveals cell-type-specific requirement for *rbbp4* in *ascl1b* progenitor survival

We conducted the same experimental approach as described above using the endogenous proneural Cre driver lines *ascl1b-2A-Cre, gcry1:GFP$^{is75}$* and *neurod1-2A-Cre, gcry1:GFP$^{is77}$*, but in combination with the *rbbp4$^{on}$* allele to test for cell-type-specific conditional inactivation (**Figure 6H–M**). Embryos from *rbbp4$^{on/+}$* + crossed with *ascl1b-2A-Cre/+; rbbp4$^{\Delta4/+}$* adults were examined for induction of gene trap RFP expression and assessed for Cre-mediated allele inversion by 5' and 3' junction genomic qPCR (**Figure 6—figure supplement 2**; **Figure 6—figure supplement 2—source data 1**). Three dpf *ascl1b-2A-Cre/+; rbbp4$^{on/\Delta4}$* larvae show RFP expression throughout the brain and neural tube and a 20% inversion rate (**Figure 6—figure supplement 2 B, B', D**). Although the larvae appeared morphologically normal, compared to controls the sectioned tissue from 2 dpf embryos showed a significant increase in activated caspase 3 labeling in the midbrain tectum and dorsal thalamic region (** p<0.01) (**Figure 6H–I and L**; **Figure 6—source data 1**). Activated caspase 3 labeling was not detected in the retina (n.s. p=0.08543), consistent with the lack of retinal expression of *ascl1b* and *Cre* (**Figure 6H"–I" and L**). Conditional inactivation of *rbbp4$^{on}$* with *neurod1-2A-Cre* also lead to induction of RFP expression in both the brain and the retina, albeit at lower levels, and 20% cassette inversion efficiency (**Figure 6—figure supplement 2 C, C', E**; **Figure 6—figure supplement 2—source data 1**), However, in contrast to conditional inactivation with *ascl1b-2A-Cre*, inactivation with *neurod1-2A-Cre* did not lead to significantly different levels of caspase 3 labeling in the 2 dpf brain (n.s. p=0.3739) or retina (n.s. p=0.6434) compared to controls (**Figure 6J–K, M**; **Figure 6—source data 1**). These results

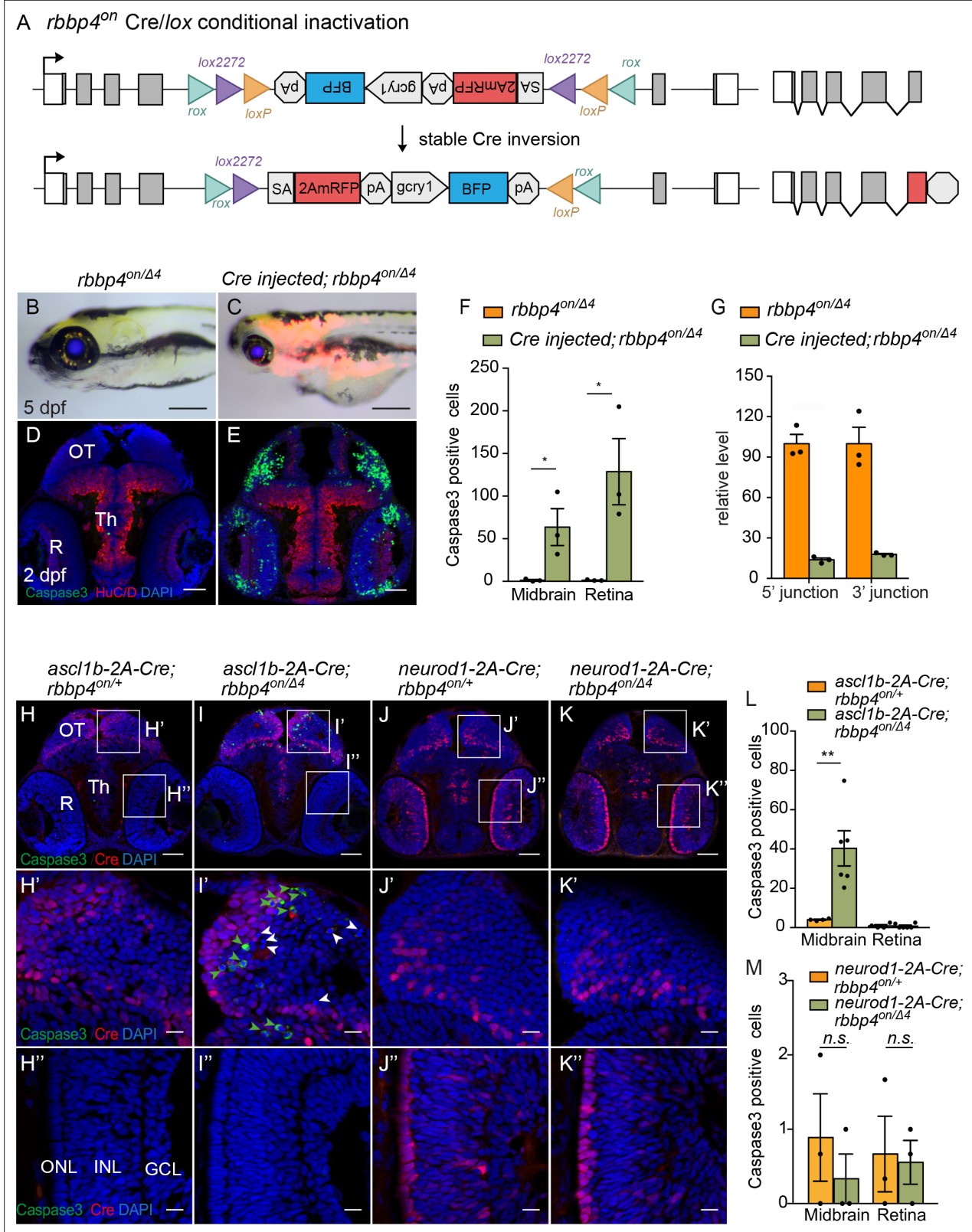

**Figure 6.** Ubiquitous and cell-type specific Cre-mediated conditional inactivation with *rbbp4-on*. (**A**) Diagram of expected Cre-mediated inversion of *rbbp4^on^* to "off" orientation. (**B**) Normal morphological phenotype in 5 dpf transheterozygous *rbbp4^on/Δ4^*larva. (**C**) Induction of microcephaly and microphthalmia and mRFP expression in Cre injected 5 dpf transheterozygous *rbbp4^on/Δ4^*larva. (**D**) Absence of activated caspase-3a labeling in sectioned tissue from 2 dpf uninjected transheterozygous *rbbp4^on/Δ4^*embryo. (**E**) Activated caspase-3a labeling in the midbrain and retina of 2

*Figure 6 continued on next page*

*Figure 6 continued*

dpf transheterozygous *rbbp4*$^{on/\Delta4}$embryo after Cre injection. (**F**) Quantification of caspase-3a labeling in control *rbbp4*$^{on/\Delta4}$(n=3) and Cre-injected *rbbp4*$^{on/\Delta4}$(n=3) midbrain (* p<0.05) and retina (* p<0.05). (**G**) Genomic DNA qPCR quantification of *rbbp4*$^{on}$ original orientation 5' and 3 junctions in control *rbbp4*$^{on}$ (n=3) and Cre injected *rbbp4*$^{on/\Delta4}$ (n=3). (**H – K"**) Activated caspase-3a and Cre labeling in sectioned head tissue from 2 dpf *ascl1b-2A-Cre; rbbp4*$^{D4/+}$ (**H-H"**), *ascl1b-2A-Cre; rbbp4*$^{on/\Delta4}$ (**I-I"**), *neurod1-2A-Cre; rbbp4*$^{\Delta4/+}$ (**J-J"**), and *neurod1-2A-Cre; rbbp4*$^{on/\Delta4}$ (**K-K"**), embryos. Green arrowheads, activated caspase-3a-positive cells. White arrowheads, hypercondensed and fragmented nuclei. (**L**) Quantification of caspase-3a labeling in *ascl1b-2A-Cre; rbbp4*$^{on/+}$ (n=4) and *ascl1b-2A-Cre; rbbp4*$^{on/\Delta4}$ (n=6) midbrain (** p<0.01) and retina (n.s. p=0.8543). (**M**) Quantification of caspase-3a labeling in *neurod1-2A-Cre; rbbp4*$^{on/+}$ (n=3) and *neurod1-2A-Cre; rbbp4*$^{on/\Delta4}$ (n=3) midbrain (n.s. p=0.3739) and retina (n.s. p=0.6433). OT, optic tectum; R, retina; Th, thalamic region. Error bars represent mean ± s.e.m. with two-tailed *t*-test. Scale bars: 200 μm (**B, C**), 50 μm (D, E, H – K). 10 μm (H' – K").

The online version of this article includes the following source data and figure supplement(s) for figure 6:

**Source data 1.** Source data for quantification of activated caspase-3a labeling and *rbbp4*$^{on}$ inversion after Cre injection, activated caspase-3a labeling by *ascl1b-2A-Cre* and *neurod1-2A-Cre*.

**Figure supplement 1.** Live imaging and molecular analysis of *rbbp4*$^{on}$ conditional inactivation by Cre injection.

**Figure supplement 2.** Induction of primary reporter RFP expression and qPCR quantification of *rbbp4*$^{on}$ allele inversion by *ascl1b-2A-Cre* and *neurod1-2A-Cre*.

**Figure supplement 2—source data 1.** Source data for qPCR quantification of *rbbp4*$^{on}$ inversion by *ascl1b-2A-Cre* and *neurod1-2A-Cre*.

suggest Cre-mediated conditional inactivation of *rbbp4* in the *ascl1b* neural progenitor population leads to apoptosis. Although *rbbp4* is expressed in the *neurod1* population as shown by induction of RFP, the absence of apoptosis indicates *rbbp4* may not be required for survival in committed progenitors or newborn neurons. These results support two critical conclusions: First, conditional inactivation of *rbbp4*$^{on}$ with ubiquitous Cre leads to efficient stable inversion of the cassette from 'on' to 'off', and replicates the *rbbp4* loss-of-function phenotype. Second, *rbbp4*$^{on}$ can be inactivated in a cell-type-specific manner that reveals a potential cell-specific function for *rbbp4* in *ascl1b*-neural progenitor cell survival.

## *rb1*$^{off}$ allele provides robust gene knockdown and loss of function

To validate the UFlip approach for generating conditional alleles at a second locus, we characterized the activity of the *rb1*$^{off\ is58}$ and *rb1*$^{on\ is57}$ alleles (**Figure 7**). RT-qPCR of homozygous *rb1*$^{off}$ 3 dpf larvae with primers spanning *rb1* exons 6–8 showed >99% knockdown of *rb1* mRNA levels in comparison to wild-type sibling larvae (**Figure 7A, B**; **Figure 7—source data 1**). To simplify identification of trans heterozygous *rb1* mutant larvae, we isolated an *rb1* exon 2 CRISPR integration allele, *rb1-stop-myl7:GFP*$^{is59}$ (*rb1-stop*) using a targeting vector design similar to our recently published method to isolate endogenous Cre lines in proneural genes (**Almeida et al., 2021**). The *stop-myl:GFP* cassette contains three translation termination codons, in each of the three reading frames, followed by a transcription terminator plus a *myl7:GFP* secondary marker (**Figure 7—figure supplement 1A, B**). The *rb1*$^{stop}$ allele is larval lethal in combination with the *rb1*$^{\Delta7}$ indel mutation and homozygous *rb1*$^{stop/stop}$ 5 dpf larvae show the same level of increased numbers of pH3 labeled cells throughout the midbrain (n.s. p=0.2951) and retina (n.s. *P*=0.1534) (**Figure 7—figure supplement 1C**; **Figure 7—figure supplement 1—source data 1**) as *rb1*$^{\Delta7/\Delta7}$ homozygotes (**Schultz et al., 2018**). Trans-heterozygous *rb1*$^{off/stop}$ larvae from a cross between *rb1*$^{off/+}$ and *rb1*$^{stop/+}$ heterozygotes are easily distinguished from siblings by the expression of lens BFP by the *rb1*$^{off}$ allele (**Figure 7D** arrowhead) and heart GFP by the *rb1*$^{stop}$ allele (**Figure 7D** asterisk). *rb1*$^{off/stop}$ 5 dpf larvae appear morphologically normal compared to wild type, with the exception of the lack of a swim bladder (**Figure 7C, D**), similar to *rb1*$^{\Delta7/stop}$ larvae (**Figure 7E**). In contrast to wild type (**Figure 7F**), 5 dpf sectioned head tissue shows pH3-labeled cells throughout the *rb1*$^{off/stop}$ midbrain (*** p<0.001) and retina (**** p<0.0001), like *rb1*$^{\Delta7/stop}$ transheterozygotes (**Figure 7F–H**; **Figure 7—figure supplement 2A**; **Figure 7—source data 1**) and *rb1*$^{stop/stop}$ and *rb1*$^{\Delta7/\Delta7}$ homozygotes (**Figure 7—figure supplement 1C**). Together these results demonstrate the *rb1*$^{off}$ allele provides robust gene knockdown and recapitulates the zebrafish *rb1* loss-of-function morphological and cellular phenotypes.

## *rb1*$^{on}$ allele maintains *rb1* function

The *rb1*$^{on}$ allele (**Figure 7I**) was examined for its effect on *rb1* mRNA expression and gene activity. RT-qPCR of *rb1*$^{on/+}$ heterozygotes and *rb1*$^{on/on}$ homozygotes showed 84% and 83% levels, respectively, of *rb1* mRNA compared to wild-type controls (**Figure 7J**; **Figure 5—source data 1**). Like wildtype,

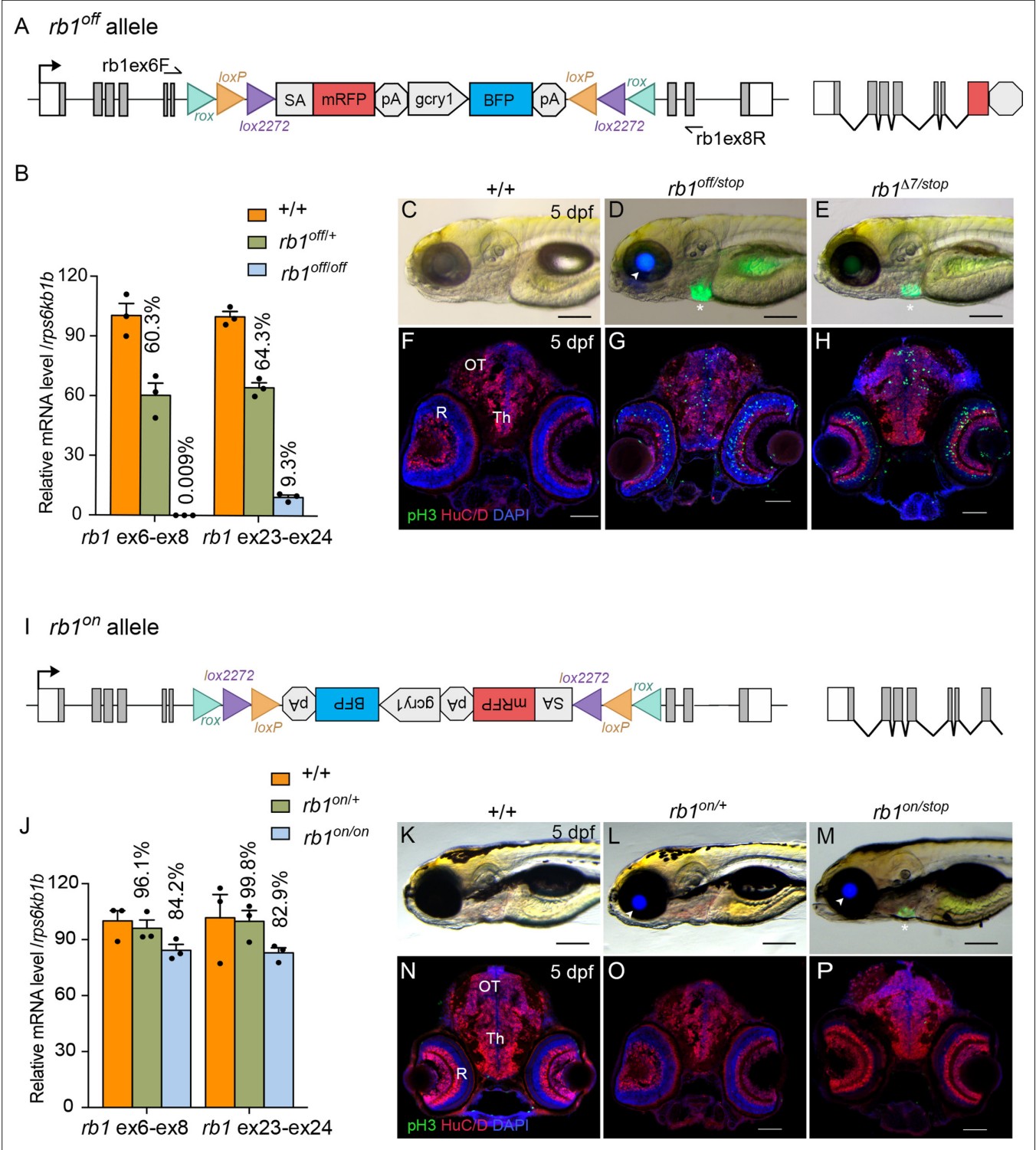

**Figure 7.** Molecular and phenotypic characterization of *rb1*<sup>off</sup> and *rb1*<sup>on</sup> alleles. (**A**) Diagram of the *rb1*<sup>off</sup> allele. (**B**) Plot of RT-qPCR results from wild type +/+ (n=3), heterozygous *rb1*<sup>off/+</sup> (n=3), and homozygous *rb1*<sup>off/off</sup> (n=3) larvae showing the relative level of *rb1* mRNA transcript using reference gene *rps6kb1b*. Primer pairs were located in exons 6 and 8, or downstream exons 23 and 24. (**C – E**) Gross phenotype of wildtype +/+ (**C**), *rb1*<sup>off/stop</sup> (**D**), and *rb1*<sup>Δ7/stop</sup> (**E**) 5 dpf larvae. Arrowhead in **D** points to *rb1*<sup>off</sup> allele *gcry1:BFP* secondary reporter expression in lens. Asterisk marks the *rb1*<sup>stop</sup> allele *mly7:GFP* secondary reporter expression in heart. (**F – H**) pH3 and HuC/D labeling of sectioned head tissue from 5 dpf +/+ (**F**), *rb1*<sup>off/stop</sup> (**G**), and *rb1*<sup>Δ7/stop</sup> (**H**). (**I**) Diagram of the *rb1*<sup>on</sup> allele. (**J**) Plot of RT-qPCR results from wild type +/+ (n=3), heterozygous *rb1*<sup>on/+</sup> (n=3), and homozygous *rb1*<sup>on/on</sup> (n=3) larvae showing the relative level of *rb1* mRNA transcript using reference gene *rps6kb1b*. Primer pairs were located in exons 6 and 8, or downstream exons 23

*Figure 7 continued on next page*

*Figure 7 continued*

and 24. (**K, L**) Gross phenotype of wildtype +/+ (**K**), heterozygous *rb1^on/+* (**L**) and transheterozygous *rb1^on/stop* (**M**) 5 dpf larvae. Arrowhead in **L, M** points to *rb1^off* allele *gcry1:BFP* secondary reporter expression in lens. Asterisk in **M** marks the *rb1^stop* allele *mly7:GFP* secondary reporter expression in heart. pH3 and HuC/D labeling of sectioned head tissue from 5 dpf +/+ (**N**), heterozygous *rb1^on/+* (**O**) and transheterozygous *rb1^on/stop* (**P**).OT, optic tectum; R, retina; Th, thalamic region. Error bars represent mean ± s.e.m. Scale bars: 200 µm (**C–E, K–M**), 50 µm (**F–H, N–P**).

The online version of this article includes the following source data and figure supplement(s) for figure 7:

**Source data 1.** Source data for *rb1* RT-quantitative PCR analysis in wildtype, heterozygous and homozygous embryos from *rb1^off/+* and *rbbp4^on/+* incrosses.

**Figure supplement 1.** The *rb1-stop* integration allele recapitulates the *rb1Δ7* indel loss-of-function phenotype.

**Figure supplement 1—source data 1.** Source data for quantification of pH3 positive cells in midbrain and retina of wildtype +/+, heterozygous *rb1^stop/stop*, homozygous *rb1^stop/stop*, and homozygous *rb1^Δ7/Δ7* 5 dpf larva.

**Figure supplement 2.** Quantification of pH3 labeled cells in *rb1^off/stop*, *rb1^Δ7/stop*, and *rb1^off/stop* mutant larvae.

**Figure supplement 2—source data 1.** Source data for quantification of pH3 positive cells in midbrain and retina of wildtype +/+, heterozygous *rb1^on/+*, and transheterozygous *rb1^on/stop* 5 dpf larva.

heterozygous *rb1^on/+* and trans-heterozygous *rb1^on/stop* 5 dpf larvae appear morphologically normal (*Figure 7K–M*). pH3 labeling in 5 dpf sectioned head tissue showed no significant difference between wild type (+/+) and *rb1^on/+* (*Figure 7N, O*; *Figure 7—figure supplement 2B*; *Figure 7—figure supplement 2—source data 1*) in the midbrain (n.s. p=0.1118) and a slightly significant increase in the retina (* p>0.05). In *rb1^on/stop* transheterozygous larvae an increase in pH3 labeled cells was detected in the midbrain (** p<0.01) and retina (** p0.01) (*Figure 7P*; *Figure 7—figure supplement 2B*; *Figure 7—source data 1*). However, the actual increase in number was minor, from 10 to ~30, in contrast to hundreds of pH3-positive cells in homozygous mutant *rb1* tissues. Together, these results demonstrate the *rb1^off* allele knocks down *rb1* mRNA expression and gene activity as expected, while the *rb1^on* allele does not significantly disrupt either.

## Cre/*lox* recombination of *rb1^off* leads to stable inversion and conditional gene rescue

To test the *rb1^off* and *rb1^on* alleles for Cre-mediated conditional rescue and inactivation, we used the same approach as outlined above for the conditional *rbbp4^off* and *rbbp4^on* alleles. First, 12.5 pg Cre mRNA injection was used to test for recombinase mediated inversion and conditional gene rescue in embryos from a cross between *rb1^off/+* and *rb1^stop/+* adults (*Figure 8A–G*). Stable inversion of the *rb1^off* cassette to the 'on' orientation was confirmed by PCR and sequence analysis, which showed the expected sequences at the 5' and 3' junctions after inversion at the *lox* sites (*Figure 8—figure supplement 1*). In comparison to uninjected *rb1^off/stop* larvae (*Figure 8B*), 5 dpf *rb1^off/stop* Cre injected larvae showed a significant reduction in the number of pH3-labeled cells in the midbrain (** p<0.01) and retina (** p<0.01) (*Figure 8C, F*; *Figure 8—source data 1*). Genomic DNA quantitative PCR of the original 5' and 3' *rb1^off* allele junction fragments in the *Cre* injected embryos indicated a>78% inversion rate (*Figure 8G*; *Figure 8—source data 1*). Together these data demonstrate Cre mediated recombination of the *rb1^off* allele leads to efficient, stable inversion of the cassette to the 'on' orientation and conditional rescue of the *rb1* loss-of-function phenotype.

## *neurod1*-specific *rb1^off* conditional rescue demonstrates *rb1* is required to suppress *neurod1* neural progenitor cell cycle entry

The proneural *neurod1-2A-Cre, gcry1:GFP* ^is77^ driver was used to test for conditional gene rescue of *rb1* in the *neurod1* progenitor population. Five dpf larvae from adult *rb1^off/+* crossed with *neurod1-2A-Cre/+; rb1^stop/+* was labeled with antibodies to Cre and pH3 (*Figure 8D–I*). In comparison to control *rb1^off/stop* larvae, *neurod1-2A-Cre/+; rb1^off/stop* larvae showed a significant reduction in pH3 positive cells in the midbrain (* p<0.05) and retina (** p<0.01) where Cre is expressed (*Figure 8D–H*; *Figure 8—source data 1*). Rescue of the phenotype correlated with >34% Cre mediated inversion of *rb1^off* to the 'on' orientation (*Figure 8I*; *Figure 8—source data 1*). The results demonstrate cell-type-specific conditional rescue and a requirement for *rb1* to suppress cell cycle entry in *neurod1* progenitors.

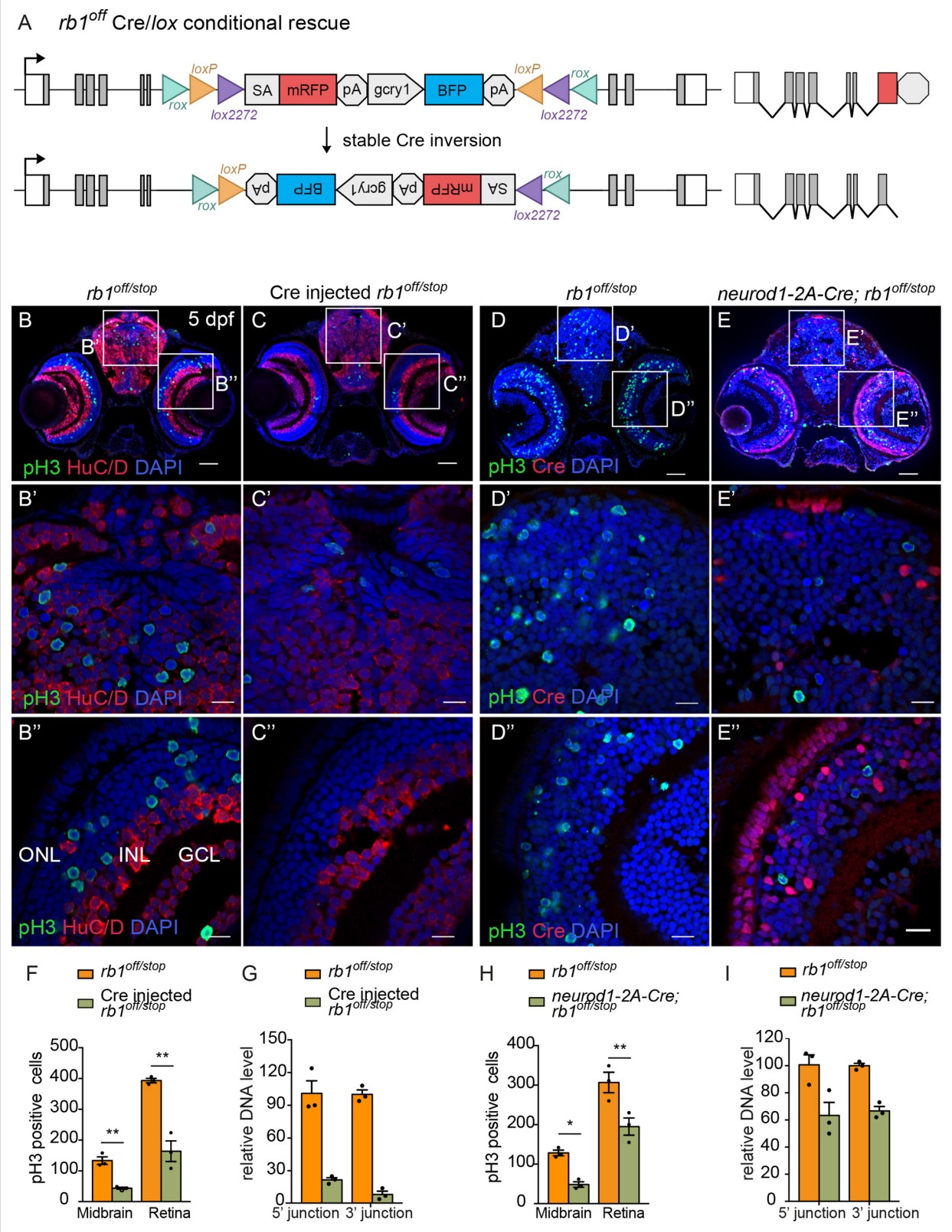

**Figure 8.** Ubiquitous and proneural *neurod1*-specific Cre-mediated conditional rescue with *rb1^off^*. (**A**) Diagram of expected Cre-mediated inversion of *rb1^off^* to on orientation. (**B, C**) pH3 and HuC/D labeling of larval sectioned head tissue from 5 dpf transheterozygous *rb1^off/stop^* (B – B") and Cre injected *rb1^off/stop^* (C – C"). (**D, E**) pH3 and Cre labeling of larval sectioned head tissue from 5 dpf transheterozygous *rb1^off/stop^* (D – D") and *neurod1-2A-Cre; rb1^off/stop^* (E – E"). (**F**) Quantification of pH3-positive cells in control *rb1^off/stop^* (n=3) and Cre injected *rb1^off/stop^* (n=3) midbrain (** p<0.01) and retina (** p<0.01).

*Figure 8 continued on next page*

*Figure 8 continued*

(**G**) Genomic DNA qPCR quantification of $rb1^{off}$ original orientation DNA 5' and 3' junctions in control $rb1^{off/stop}$ (n=3) and Cre injected $rb1^{off/stop}$ (n=3). (**H**) Quantification of pH3-positive cells in $rb1^{off/stop}$ (n=3) and $neurod1$-$2A$-$Cre$; $rb1^{off/stop}$ (n=3) midbrain (** p<0.01) and retina (* p<0.05). (**I**) Genomic DNA qPCR quantification of $rb1^{off}$ original orientation DNA 5' and 3' junctions in control $rb1^{off}$ (n=3) and $neurod1$-$2A$-$Cre$; $rb1^{off/stop}$ (n=3). Error bars represent mean ± s.e.m. with two-tailed *t*-test. Scale bars: 50 µm (B - E), 10 µm (B' – E").

The online version of this article includes the following source data and figure supplement(s) for figure 8:

**Source data 1.** Source data for quantification of pH3 labeling and $rb1^{off}$ inversion after Cre injection and $neurod1$-$2A$-$Cre$.

**Figure supplement 1.** Molecular analysis of $rb1^{off}$ inversion and conditional rescue by Cre injection.

## Cre/*lox* recombination of $rb1^{on}$ leads to efficient conditional gene inactivation and recapitulates cellular $rb1$ loss-of-function phenotype

Conditional rescue using the $rb1^{on}$ allele was first tested by injection of 12.5 pg *Cre* mRNA into embryos from a cross between heterozygous $rb1^{on/+}$ and $rb1^{stop/+}$ adults to induce inversion from the passive to the active orientation (**Figure 9A**). At 3 dpf $rb1^{on/stop}$ uninjected and injected larvae appear morphologically normal like control $rb1^{on/+}$ larvae (**Figure 9—figure supplement 1 A, B**). However, in comparison to uninjected control sectioned head tissue (**Figure 9B**), Cre injected $rb1^{on/stop}$ showed increased levels of pH3 labeling throughout the brain and retina (**Figure 9C**), similar to the loss-of-function phenotype of $rb1^{\Delta 7/stop}$ and $rb1^{off/stop}$ (**Figure 7D and E**). Quantification showed a significant increase in the number of pH3-positive cells in injected larvae in both the midbrain (**** p<0.0001) and retina (** p<0.01) (**Figure 9F**; **Figure 9—source data 1**). Induction of the mutant phenotype correlated with >80% inversion rate by Cre recombination (**Figure 9G**; **Figure 9—source data 1**). Stable inversion of the UFlip cassette in *Cre* injected embryos was confirmed by PCR junction analysis and sequencing of the junction amplicons (**Figure 9—figure supplement 1 C–F**). These results demonstrate Cre recombination leads to stable inversion of the UFlip cassette and conditional gene inactivation which replicates the $rb1$ loss-of-function phenotype at the cellular level.

## $neurod1$-specific $rb1^{on}$ conditional inactivation leads to neural progenitor cell cycle entry

The proneural $neurod1$-$2A$-$Cre$, $gcry1$:$GFP^{is77}$ driver was used to test for conditional inactivation of $rb1$ in the $neurod1$ progenitor population. 3 dpf larvae from adult $rb1^{on/+}$ crossed with $neurod1$-$2A$-$Cre/+$; $rb1^{stop/+}$ was labeled with antibodies to Cre and pH3 (**Figure 9D–I**). In comparison to control $rb1^{on/stop}$ larvae (**Figure 9D**), $neurod1$-$2A$-$Cre/+$; $rb1^{on/stop}$ larvae showed a significant increase in pH3 positive cells in the midbrain (*** p<0.001) and retina (* p<0.05) where Cre is expressed (**Figure 9E and H**; **Figure 9—source data 1**). Induction of the mutant phenotype correlated with >25% inversion of $rb1^{on}$ to the 'off' orientation (**Figure 9I**; **Figure 9—source data 1**). The results are consistent with the $rb1^{off}$ conditional rescue experiment, and demonstrate a requirement for $rb1$ in suppressing cell cycle entry in the $neurod1$ progenitor cell population in the developing midbrain and retina.

Overall, these analyses show conditional alleles generated by targeted integration of the floxed UFlip cassette provide robust control of gene activity. Combined with cell-type-specific Cre drivers, the alleles demonstrate cell-type-specific requirements for gene activity through both conditional inactivation and rescue.

## Discussion

In this study, we used GeneWeld CRISPR-Cas9 targeting to isolate Cre/*lox* regulated conditional alleles with linked reporters by integration of a floxed gene trap cassette, UFlip, at a single gRNA site in an intron. Gene 'off' and gene 'on' germline UFlip alleles were recovered for two genes, *rbbp4* and *rb1*. Our results showed the UFlip cassette led to >99% gene knockdown at the molecular level, did not disrupt gene activity in the passive orientation, allowed stable Cre/*lox* mediated inversion for conditional inactivation or rescue, and recapitulated expected loss-of-function phenotypes at the morphological and cellular levels. We further demonstrated using proneural Cre drivers that conditional gene inactivation or rescue can be restricted to distinct neural progenitor cell populations. These studies revealed cell-type-specific requirements for gene activity that had previously only been inferred by global loss-of-function homozygous mutant analysis. Our approach is simple in design,

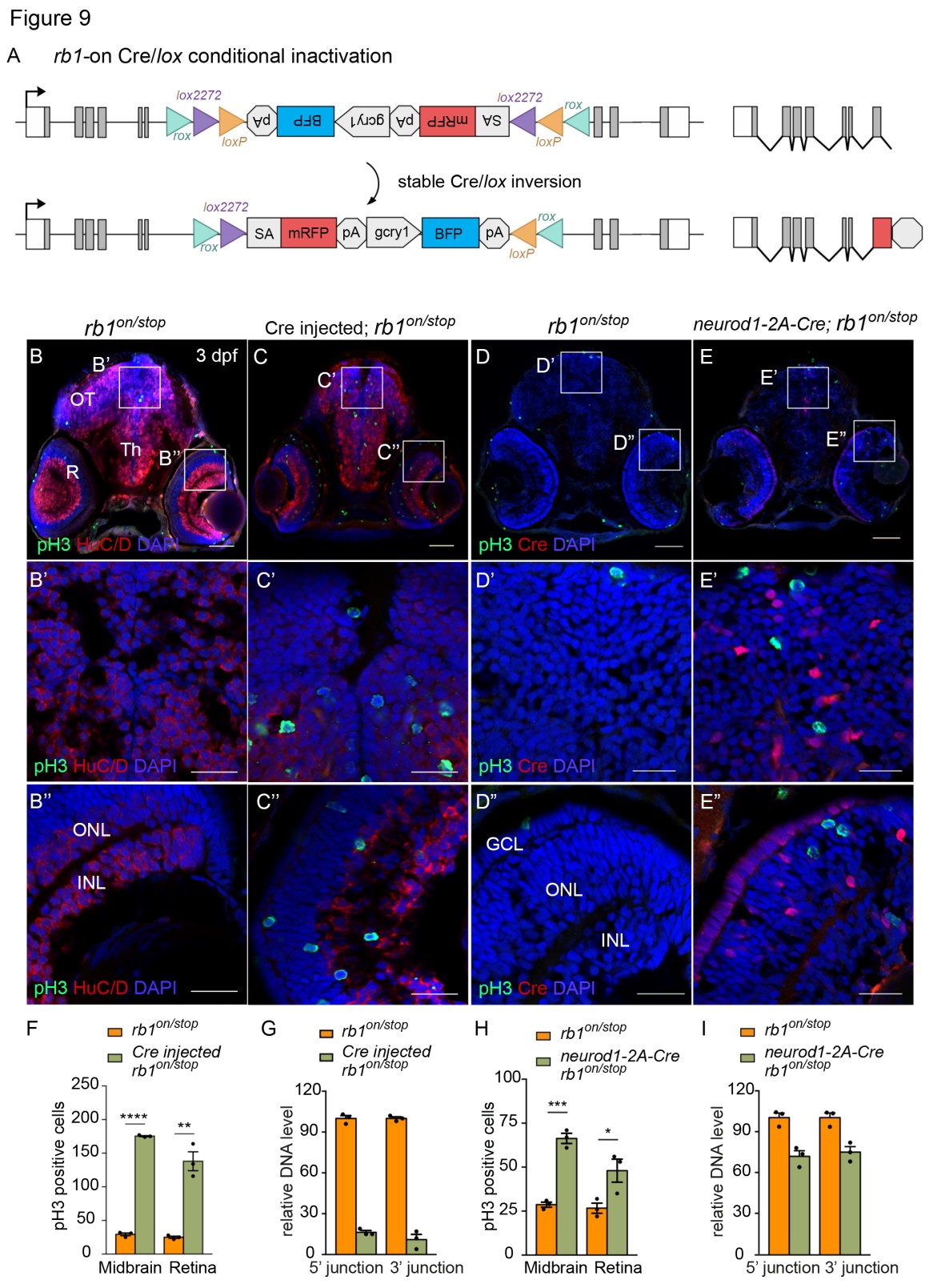

**Figure 9.** Ubiquitous and proneural *neurod1*-specific Cre-mediated conditional inactivation with *rb1-on*. (**A**) Diagram of expected Cre mediated inversion of *rb1-on* to 'off' orientation. (**B, C**) pH3 and HuC/D labeling of larval sectioned head tissue from 5 dpf transheterozygous *rb1^on/stop^* (B – B") and Cre injected *rb1^on/stop^* (C – C"). (**D, E**) pH3 and Cre labeling of larval sectioned head tissue from 5 dpf transheterozygous *rb1^on/stop^* (D – D") and *neurod1-2A-Cre; rb1^on/stop^* (E – E"). (**F**) Quantification of pH3-positive cells in control *rb1^on/stop^* (n=3) and Cre *rb1^on/stop^* (n=3) injected midbrain (**** p<0.0001) and

*Figure 9 continued on next page*

*Figure 9 continued*

retina (** p<0.01). (**G**) Genomic DNA qPCR quantification of *rb1*$^{on}$ original orientation DNA 5' and 3' junctions in control *rb1*$^{on/stop}$ (n=3) and Cre injected *rb1*$^{on/stop}$ (n=3). (**H**) Quantification of pH3-positive cells in control *rb1*$^{on/stop}$ (n=3) and *neurod1-2A-Cre; rb1*$^{on/stop}$ (n=3) midbrain (*** p<0.001) and retina (* p<0.05). (**I**) Genomic DNA qPCR quantification of *rb1*$^{on}$ original orientation DNA 5' and 3' junctions in control *rb1*$^{on/stop}$ (n=3) control and *neurod1-2A-Cre; rb1*$^{on/stop}$ (n=3). Error bars represent mean ± s.e.m. with two-tailed *t*-test. Scale bars: 50 µm (B - E), 20 µm (B' – E").

The online version of this article includes the following source data and figure supplement(s) for figure 9:

**Source data 1.** Source data for quantification of pH3 labeling and *rb1*$^{on}$ inversion after Cre injection and *neurod1-2A-Cre*.

**Figure supplement 1.** Molecular analysis of *rb1-on* inversion and conditional inactivation by Cre injection.

efficient, generates effective Cre/*lox* regulated alleles for genetic mosaic analysis and can be applied to any gene of interest.

We assembled UFlip vectors with 48 bp homology arms to generate integrations in *rbbp4* and *rb1* in both the 'off' and 'on' orientations. Of the four potential targeted integration alleles, we recovered three precise alleles at a frequency of 4–14% of founders screened. This frequency is lower than our previous reports using GeneWeld to knock in fluorescent reporters (22–100%) (*Wierson et al., 2020*) or 2A-Cre (10–100%) (*Almeida et al., 2021*; *Wierson et al., 2020*). In those studies, injected embryos showing expression of the fluorescent tag, or Cre-mediated switching of a floxed reporter, were selected and raised to adulthood to screen for germline transmission. In this report, some lines were generated with vectors that only enabled embryo selection based on expression of the secondary reporter, which does not reflect on target integration. In support of this, only 7 and 8 adults needed to be screened to recover the *rbbp4*$^{off}$ and *rb1*$^{off}$ 2A-mRFP UFlip alleles, respectively, which showed mRFP expression in the injected embryos. This is in comparison to 25 adults that were screened to recover the original *rb1*$^{on}$ mRFP UFlip alleles. Moreover, starting with an allele in one orientation, it was straightforward to use Dre recombination at *rox* sites flanking the UFlip cassette to recover an allele in the opposite orientation. Dre/*rox* recombination has been shown to be highly efficient in somatic tissue (*Park and Leach, 2013*). By injection of *Dre* mRNA into *rbbp4*$^{off}$ embryos we readily recovered the *rbbp4*$^{on}$ allele in the next generation from 7/11 adults, demonstrating efficient Dre recombinase activity in the zebrafish germline. Previously reported methods for recovery of intron targeted floxed conditional alleles varied widely in frequency, from 12% for homology directed integration from a linear template with long 1–1.5 kb homology arms (*Hoshijima et al., 2016*), 53% by homologous recombination from a circular plasmid with 1 kb long homology arms (*Sugimoto et al., 2017*), to 56–60% for integration by Non-Homologous End Joining after preselection of a primary reporter (*Han et al., 2021*; *Li et al., 2019*). The simplicity of UFlip vector construction and ability to recover precise integrations, in either orientation, underscore the power of our approach for isolation of stable Cre/*lox* regulated conditional alleles.

Similar to our previous findings when isolating endogenous Cre drivers (*Almeida et al., 2021*), the UFlip founders transmitted both precise and imprecise integration events. In some cases, the imprecise event was a simple homology arm duplication or short insertion at either the 5' or 3' junction, which would be predicted not to affect gene function since the integrated cassette is located in an intron. As predicted, the *rb1*$^{off}$ allele, from which we were able to establish a line, had a 5' homology arm duplication separated by a 115 bp insertion. This did not affect the ability of the cassette to efficiently invert in response to Cre recombinase. When inverted into the passive orientation the cassette allowed effective conditional rescue, indicating the additional sequences in the intron flanking the cassette didn't disrupt splicing or *rb1* gene activity.

For effective conditional gene inactivation, it is essential that the method leads to robust knockdown of wildtype mRNA expression and recapitulates loss-of-function phenotypes for genetic analysis. Our UFlip conditional construct is based on the Tol2 transposon RP2 gene trap, which contains a transcriptional terminator that effectively blocks transcription readthrough and reduces transcript levels by >99% (*Clark et al., 2011*; *Ichino et al., 2020*). The efficacy of gene knockdown by the UFlip gene trap was demonstrated by quantitative RT-qPCR and revealed >99% reduction in mRNA levels. Cre/*lox*-mediated inversion of the UFlip cassette from the passive to active orientation was highly efficient after *Cre* mRNA injection or exposure to a transgenic source of Cre, generating expected phenotypes in combination with loss-of-function mutations. The recently published Cre-regulated CRISPR indel targeting study provides a rapid method for conditional mutagenesis and phenotypic knockout in embryos (*Hans et al., 2021*). In contrast, a significant advantage of our stable conditional UFlip

alleles rests on the strength of transcriptional termination by the UFlip cassette in the active orientation. In addition, Cre induction of conditional inactivation leads to mRFP fluorescent cell labeling, as described in other recent studies (*Han et al., 2021*; *Hans et al., 2021*; *Li et al., 2019*), which is critical for lineage tracing and capturing mutant cells for downstream genomics applications.

We further validated effective UFlip conditional gene inactivation at the cellular level. We previously demonstrated that *rb1* is required to prevent neural progenitor cell cycle re-entry in the developing zebrafish brain and *rbbp4* is necessary for neural progenitor survival and differentiation (*Schultz et al., 2018*). Using our *ascl1b-2A-Cre* and *neurod1-2A-Cre* drivers in combination with *rbbp4*[on] and *rb1*[on] conditional alleles, we demonstrated the requirement for each gene could be examined specifically in the *ascl1b* or *neurod1* neural progenitor populations. Rb1 is required to prevent cell cycle re-entry in *ascl1b* and *neurod1* neural progenitors. Rbbp4 appears to be required for neural progenitor survival only in *ascl1b* progenitors. Further analysis with tamoxifen regulated CreERT2 drivers will refine our understanding of the requirement for these genes in neural progenitor survival and proliferation. These conditional genetic tools provide a foundation for future mechanistic studies investigating the interaction of *rb1* and *rbbp4* specifically in neural progenitor cell cycle regulation and differentiation but can also be used with other cell-type-specific Cre drivers. Our results illustrate the robustness of the UFlip conditional alleles for cell type-specific gene knockout analysis and their potential application in different developmental or disease model contexts.

In summary, building on our previous streamlined GeneWeld approach for CRISPR-Cas short homology directed targeted integration, we describe an effective method for generating zebrafish floxed conditional alleles. We present a thorough molecular and phenotypic validation of the robustness of the UFlip floxed gene trap cassette for effective knockdown of gene expression and Cre/*lox* regulated conditional inactivation and rescue. UFlip alleles can be combined with any available cell or tissue-specific Cre driver for spatial and temporal gene knock out, enabling the investigation of gene function in specific developmental and post embryonic stages or disease models. The UFlip vector can be adapted to introduce a variety of gene modification cassettes, such as alternative exons and complete cDNAs. In contrast to Tol2 transposon binary UAS overexpression systems, targeted integration leads to control by endogenous gene regulatory elements and should increase phenotype consistency. The UFlip integration approach provides a powerful platform for generating new Cre/*lox* genetic tools to modify gene function at endogenous loci in zebrafish.

## Materials and methods

**Key resources table**

| Reagent type (species) or resource | Designation | Source or reference | Identifiers | Additional information |
|---|---|---|---|---|
| Gene (*Danio rerio*) | *ascl1b* | ensemble | ENSDARG00000009702 | |
| Gene (*Danio rerio*) | *neurod1* | ensemble | ENSDARG00000019566 | |
| Gene (*Danio rerio*) | *rb1* | ensemble | ENSDARG00000006782 | |
| Gene (*Danio rerio*) | *rbbp4* | ensemble | ENSDARG00000029058 | |
| Strain, strain background (*Danio rerio*) | WIK | Zebrafish International Resource Center | ZIRC:ZL84 | Wildtype strain of zebrafish |
| Genetic reagent (*Danio rerio*) | Tg(*ascl1b-2A-Cre*) | McGrail lab | Tg(*ascl1b-2A-Cre; gcry1:EGFP*)[is75] | Maintained in the lab of M. McGrail (*Almeida et al., 2021*) |
| Genetic reagent (*Danio rerio*) | Tg(*neurod1-2A-Cre*) | McGrail lab | Tg(*neurod1-2A-Cre; gcry1:EGFP*)[is77] | Maintained in the lab of M. McGrail (*Almeida et al., 2021*) |
| Genetic reagent (*Danio rerio*) | *rb1Δ7* | McGrail lab | *rb1Δ7*[is54] | Maintained in the lab of M. McGrail (*Solin et al., 2015*) |
| Genetic reagent (*Danio rerio*) | *rbbp4Δ4* | McGrail lab | *rbbp4Δ4*[is60] | Maintained in the lab of M. McGrail (*Schultz et al., 2018*) |
| Genetic reagent (*Danio rerio*) | Tg(*rb1-UFlip-on*) | This paper | Tg(*rb1-i6-UFlip-rox-lox2272-loxP-inverted<RFP;gcry1:BFP >lox2272-loxP-rox-on*)[is57] | Available from M. McGrail lab |

*Continued on next page*

*Continued*

| Reagent type (species) or resource | Designation | Source or reference | Identifiers | Additional information |
|---|---|---|---|---|
| Genetic reagent (*Danio rerio*) | *Tg(rb1-UFlip-off)* | This paper | *Tg(rb1-i6-UFlip-rox-loxP-lox2272<RFP;gcry1:BFP >loxP-lox2272 -rox-off)^is58* | Available from M. McGrail lab |
| Genetic reagent (*Danio rerio*) | *Tg(rb1-UFlip-off)* | This paper | *Tg(rb1-i6-UFlip-rox-loxP-lox2272<2A-RFP;gcry1:BFP >loxP-lox2272 -rox-off)^is63* | Available from M. McGrail lab |
| Genetic reagent (*Danio rerio*) | *Tg(rbbp4-UFlip-off)* | This paper | *Tg(rbbp4-i4-UFlip-rox-lox2272-loxP-<2A-RFP;gcry1:BFP >lox2272-loxP-rox-off)^is61* | Available from M. McGrail lab |
| Genetic reagent (*Danio rerio*) | *Tg(rbbp4-UFlip-on)* | This paper | *Tg(rbbp4-i4-UFlip-rox-lox2272-loxP-<2A-RFP;gcry1:BFP >lox2272-loxP-rox-on)^is62* | Available from M. McGrail lab |
| Genetic reagent (*Danio rerio*) | *Tg(rb1-stop-PRISM)* | This paper | *Tg(rb1-3XSTOP; myl7:GFP)^is59* | Available from M. McGrail lab |
| Recombinant DNA reagent | pT3TS-nCas9n | Wenbiao Chen | Addgene:46,757 | Plasmid for in vitro synthesis of Cas9 mRNA |
| Recombinant DNA reagent | pT3TS-Cre | Karl Clark | | Plasmid for in vitro synthesis of Cre mRNA, available from K. Clark lab (*Clark et al., 2011*) |
| Recombinant DNA reagent | pT3TS-Dre | Karl Clark | | Dre cDNA (*Anastassiadis et al., 2009*) expression vector for in vitro mRNA synthesis, available from K. Clark lab |
| Recombinant DNA reagent | p-CS2-KalTA4 | Martin Distel | | Available from M. Distel lab (*Distel et al., 2009*) |
| Recombinant DNA reagent | pUFlip-floxed-mRFP, gcry1:BFP | This paper | *pUFlip(UgRNA-rox-loxP-lox2272<RFP; gcry1:BFP >loxP-lox2272-rox-UgRNA)* | decommissioned |
| Recombinant DNA reagent | pUFlip-floxed-mRFP, myl7:BFP | This paper | *pUFlip(UgRNA-rox-loxP-lox2272<RFP; myl7:BFP >loxP-lox2272-rox-UgRNA)* | decommissioned |
| Recombinant DNA reagent | pUFlip-floxed-2A-mRFP, gcry1:BFP | This paper | *pUFlip(UgRNA-rox-loxP-lox2272<2A-RFP; gcry1:BFP >loxP-lox2272-rox-UgRNA)* | available from M. McGrail lab; Deposited at Addgene |
| Recombinant DNA reagent | pUFlip-floxed-2A-KalTA4, gcry1:BFP | This paper | *pUFlip(UgRNA-rox-loxP-lox2272<2A-KalTA4; gcry1:BFP >loxP-lox2272-rox-UgRNA)* | available from M. McGrail lab; Deposited at Addgene |
| Recombinant DNA reagent | pSTOP-PRISM-3Xstop-myl7:GFP | This paper | *pPRISM(UgRNA-3XSTOP; myl7:GFP-UgRNA)* | available from J. Essner lab; Deposited at Addgene |
| Sequence-based reagent | | This paper | PCR primers and oligos | See *Table 1* |
| Commercial assay or kit | NEBuilder HiFi DNA Assembly Cloning kit | New England Biolabs | Catalog # E5520S | |
| Commercial assay or kit | PureYield Plasmid Miniprep System | Promega | Catalog # A1223 | |
| Commercial assay or kit | mMessage mMACHINE T3 Transcription Kit | Ambion | Catalog # AM1348 | |
| Commercial assay or kit | RNA Clean and Concentrator Kit (RCC) | Zymo | Catalog # R1013 | |
| Commercial assay or kit | pCR4 TOPO TA Cloning Kit | ThermoFisher/ Invitrogen | ThermoFisher:K457502 | |
| Commercial assay or kit | Superscript III | Invitrogen | Catalog # 18080093 | |
| Commercial reagent | SYBR Green | BioRad | Catalog # 1725271 | |
| Commercial reagent | Tissue-Tek O.C.T. Compound | Fisher | Catalog # 4,583 | |
| Software, algorithm | ICE | Synthego | Inference of CRISPR Edits (ICE) https://www.synthego.com/products/bioinformatics/crispr-analysis | Indel analysis of Sanger sequenced DNA |
| Software | Graphpad PRISM | | | Statistical analyses |

*Continued on next page*

*Continued*

| Reagent type (species) or resource | Designation | Source or reference | Identifiers | Additional information |
|---|---|---|---|---|
| Antibody | Rabbit polyclonal anti -Caspase-3a | BD Pharmingen | Catalog # 559,565 | (1:500) |
| Antibody | Mouse monoclonal anti-Cre Recombinase | Chemicon | Catalog # MAB3120 | (1:500) |
| Antibody | Mouse monoclonal anti-HuC/D | Invitrogen | Catalog # A21271 | (1:500) |
| Antibody | Mouse monoclonal anti-phospho-Histone H3 (Ser10) | Millipore | Catalog # 05–806 | (1:500) |
| Antibody | Rabbit polyclonal anti-phospho-Histone H3 | Millipore | Catalog # 06–570? | (1:1000) |
| Antibody | Rabbit polyclonal anti-Rbbp4 | Bethyl Laboratories | Catalog # A301-206A | (1:200) |
| Chemical compound, drug | DAPI | Invitrogen | Catalog # D1306 | (0.5 ug/ml) |

## Ethics declarations and approval for animal experiments

Use of zebrafish for research in this study was performed according to the Guidelines for Ethical Conduct in the Care and Use of Animals (*APA, 1986*), and carried out in accordance with Iowa State University Animal Care and Use Committee IACUC-18–279 and IACUC-20–058 approved protocols. All methods involving zebrafish were in compliance with the American Veterinary Medical Association (2020), ARRIVE (*Percie du Sert et al., 2020*) and NIH guidelines for the humane use of animals in research.

## Zebrafish strains and maintenance

Zebrafish (*Danio rerio*) were maintained on an Aquaneering aquaculture system at 26 °C on a 14 hr light/10 hr dark cycle. The WIK strain of wild type zebrafish was obtained from the Zebrafish International Resource Center (https://zebrafish.org/home/guide.php). Other zebrafish lines used in this study were previously described: *Tg(ascl1b-2A-Cre; gcry1:EGFP)*[is75] and *Tg(neurod1-2A-Cre; gcry1:EGFP)*[is77] (*Almeida et al., 2021*); *rb1Δ7*[is54] (*Solin et al., 2015*); *rbbp4Δ4*[is60] (*Schultz et al., 2018*).

## Contact for reagent and resource sharing

Further information and requests for resources and reagents should be directed to Maura McGrail (mmcgrail@iastate.edu).

## Floxed UFlip vector for CRISPR-targeted integration to generate conditional alleles

The UFlip version 1.0 floxed mRFP gene trap vector for targeted integration (*Figure 1A*) was designed to be compatible with our previously published GeneWeld CRISPR-Cas9 targeted integration strategy (*Wierson et al., 2020*). The UFlip cassette was assembled in the pPRISM parent vector (*Almeida et al., 2021*) which has Universal UgRNA sequences on and BfuAI and BspQI type II restriction enzyme sites for cloning 5' and 3' homology arms on either side of the cassette, as previously described (*Wierson et al., 2020*). In the UFlip vector, the homology arm cloning sites flank head to head oriented *rox* sites that sit outside alternating pairs of head to head *loxP* and *lox2272* sites for stable inversion after Cre mediated recombination (*Robles-Oteiza et al., 2015*; *Schnütgen et al., 2003*). Internal to the *rox* and *lox* sequences is the RP2 gene trap (*Clark et al., 2011*), consisting of a gene trap Splice Acceptor-mRFP-ocean pout (*Zoarces americanus*) antifreeze gene transcriptional termination and polyadenylation sequence (*Gibbs and Schmale, 2000*), followed by a tissue-specific lens *gcry1:BFP* or heart *myl7:BFP* secondary reporter. To assemble UFlip an intermediate vector was created by adding *Eag*I and *Cla*I restriction sites to the pPRISM backbone by PCR. Six pairs of oligos, each containing the sequence of one of the *rox*, *loxP* or *lox2272* sites, with complementary overhangs were ligated into the *Eag*I and *Cla*I digested backbone. All three reading frames of the RP2 gene trap

plus secondary reporter was directionally cloned into *Avr*II and *Sac*I restriction sites located inside the pairs of *rox/loxP/lox2272* sites. The UFlip version 2.0 vector was generated by mRFP primary reporter in the first version of the UFlipv1.0 vector was subsequently replaced with the porcine teschovirus-1 polyprotein 2 A peptide fused to mRFP. A UFlip version with a gene trap primary reporter containing the 2A-KalTA4 cDNA (*Distel et al., 2009*) for signal amplification using with a UAS:reporter is also available.

### Intronic sgRNA target site selection, UFlip homology arm design and targeted integration

To identify CRISPR gRNA sites in introns, intronic sequences from four adult female and four adult make WIK fish were amplified from fin clip DNA using the proofreading enzyme KOD and sequenced to identify non-repetitive sequences that were shared among adults in the population. Gene specific and vector Universal UgRNA synthetic gRNAs with 2'-O-Methyl at first and last bases, 3' phosphoro-thioate bonds between first 3 and last 2 bases, were ordered from Synthego . gRNA efficiency was determined by co-injection of 25 pg gRNA plus 300 pg Cas9 mRNA into one-cell stage embryos, followed by PCR amplification of the targeted intron and analysis of heteroduplex formation by gel electrophoresis. Amplicons were Sanger sequenced and the sequences analyzed for indel efficiency using Synthego's Inference of CRISPR Edits (ICE) analysis software (https://ice.synthego.com/#/). The UFlip targeting vectors were built following the GeneWeld protocol (*Welker et al., 2021*). 48 bp 5' and 3' homology arms were designed to sequences flanking the genome intronic CRISPR-Cas9 target site. The homology arms were assembled by annealing complementary oligonucleotides with appropriate overhangs for cloning into the *Bfu*AI and *Bsp*QI type IIS restriction enzyme sites that flank the UFlip cassette. To generate an active, gene 'off' allele, the 5' and 3' homology arms were cloned into the UFlip *Bfu*AI and *Bsp*QI sites, respectively. To integrate the UFlip cassette in the passive, gene 'on' orientation, the position of the cloned 5' and 3' homology arms was reversed, with the 3' homology arm cloned into the *Bfu*AI site upstream of the cassette, and the 5' homology arm cloned in the *Bsp*QI site downstream of the cassette. gRNA and homology arm oligonucleotide sequences are listed in *Table 1*.

For synthesis of *Cas9* mRNA the expression vector pT3TS-nCas9n (Addgene #46757) (*Jao et al., 2013*) was linearized with *Xba* I (New England Biolabs, R0145S). One µg linearized vector was purified with the PureYield Plasmid Miniprep System (Promega, A1223) and used as template for in vitro synthesis of capped mRNA with the Ambion mMessage Machine T3 Transcription Kit (Thermo Fisher, AM1348). in vitro synthesized mRNA was purified with the RNA Clean and Concentrator Kit RCC (Zymo, R1013).

To target CRISPR-Cas9 driven integration of the UFlip cassette into intronic sites, a 2 nl volume containing 25 pg of genomic gRNA, 25 pg of UgRNA, 10 pg of UFlip targeting vector, and 300 pg Cas9 mRNA was co-injected into one-cell stage embryos from crosses between sequence validated adult fish. Larvae were screened at 3 dpf for expression of the lens *gcry1:BFP* or heart *myl7:BFP* secondary marker. Three to four BFP-positive embryos were selected for genomic DNA isolation and confirmation of on target integration by PCR sequence analysis of 5' and 3' junctions. BFP-positive sibling embryos were raised to adulthood. Primers, gRNA and homology arm oligonucleotide sequences are listed in *Table 1* and *Table 4*.

### Isolation of stable zebrafish *rbbp4* and *rb1* UFlip and stop-PRISM integration alleles

To identify founder fish transmitting a UFlip allele, adults were outcrossed to wild-type WIK and embryos were screened for expression of the UFlip secondary reporter expressing BFP in the lens (*gcry1:BFP*) or heart (*myl7:BFP*). To identify on target integration alleles, genomic DNA was extracted from individual BFP-positive embryos by digestion in 50 mM NaOH at 95 °C for 30 min and neutralization by addition of 1/10th volume 1 M Tris-HCl pH 8.0. Genomic DNA/UFlip cassette 5' and 3' junctions were amplified by PCR with gene specific and UFlip primers listed in *Table 4*, followed by direct Sanger sequencing. BFP-positive sibling embryos from a founder that was transmitting a precise UFlip integration allele were raised to adulthood. F1 adult animals were fin clipped and the genotype of individuals confirmed by PCR and sequencing. Confirmed F1 adults were outcrossed to wild-type

**Table 4.** Primer oligonucleotide sequences.

| Primer name | Sequence | Purpose |
| --- | --- | --- |
| rbbp4e × 4 F | ACCAAACACCCCTCCAAACCAG | intron sequencing, sgRNA analysis and genome/vector 5' junction analysis |
| rbbp4e × 5 R | AGTGCACTCTCCAGAGGGGT | intron sequencing, sgRNA analysis and genome/vector 3' junction analysis |
| rb1e × 6 F | CATGTTCCTCCTGGCCAAG | genome/vector 5' junction analysis |
| rb1e × 7 R | CACAAGGTCATCTTCCATCTG | genome/vector 3' junction analysis |
| rb1e × 2 F | GAGGAGCTCCAGTCCACTAAC | genotyping and genome/vector 5' junction analysis |
| rb1e × 2 R | CCCAAAACACAAGTGCGGTAA | genotyping and genome/vector 3' junction analysis |
| R-5'-junc-Stop-pPRISM | CGGTGGCTGAGACTTAATTACT | stop-PRISM genome/vector 5' junction analysis |
| F-3'-junc-(all)-pPRISM | TTCAGATCAATTAACCCTCACC | stop-PRISM genome/vector 3' junction analysis |
| RFPj | ATGACGTCCTCGGAGGAGGC | UFlip genome/vector junction analysis |
| RFPj2 | CCTTGGTCACCTTCAGCTTG | UFlip genome/vector junction analysis |
| BactinpAj | GCAAACGGCCTTAACTTTCC | UFlip genome/vector junction analysis |
| 2AR | CATAGGACCGGGGTTTTCTT | UFlip and 2A-Cre genome/vector junction analysis |
| rb1e × 6 F | CAGCTGGACCATGTTCCTCC | QPCR |
| rb1e × 8 R | CCCTGATTACGGCGTGATGT | QPCR |
| rbbp4e × 4 F | TAGTGACGTGCTGGTCTTTG | QPCR |
| rppb4e × 5 R | CAGGACAGACCATAACCTTCTT | QPCR |
| rbbp4e × 11_1 F | CTCTGTGTCTGAGGACAACATC | QPCR |
| rbbp4e × 12_1 R | TATCCCTGAACCTCAGTGTCT | QPCR |
| rps6kb1b_1 F | TCCTGATGACTCCACACTGA | QPCR |
| rps6kb1b_1 R | GGCGAGGTGAACGGATTT | QPCR |
| BFP_F | CTGCCTCATCTACAACGTCAA | genotyping |
| BFP_R | CTTAGCGGGTTTCTTGGATCTAT | genotyping |
| neujF | TCCAACTGAACCCCAGAACT | genotyping |
| ascjF | GTCAACATGGGCTTCCAGAC | genotyping |
| rbbp4e × 2 F | GCGTGATGACAGATCTCATATTGTTTTCCC | genotyping |
| rbbp4e × 2 R | CTGGTGACATCTGGCAACCACT | genotyping |

WIK, and F2 adults again were confirmed for the presence of a precise UFlip integration allele. Individual confirmed F2 adults were outcrossed to WIK to establish independent transgenic lines.

The *rb1-stop-PRISM-myl7:GFP* allele was generated with a pPRISM (PRecise Integration with Secondary Marker) GeneWeld targeted integration vector containing a cassette with splice acceptor followed by three copies of TGA, TAA, TAG, the ocean pout (*Zoarces americanus*) antifreeze gene transcriptional termination and polyadenylation sequence (*Gibbs and Schmale, 2000*), and a *myl7:eGFP-βactin* polyadenylation secondary reporter. Previously described 5' and 3' homology arms complementary to the *rb1* exon 2 CRISPR target site (*Schultz et al., 2018*; *Wierson et al., 2020*) were cloned into the *Bfu*AI and *Bsp*QI type IIS restriction enzyme sites flanking the *stop-PRISM-myl7:eGFP* cassette. 25 pg of *rb1* exon 2 genomic gRNA, 25 pg of UgRNA, 10 pg of *rb1-stop-PRISM* targeting vector, and 300 pg *Cas9* mRNA were coinjected into one-cell stage embryos in a volume of 2 nl, and adult founders screened for transmission of the *rb1-stop-PRISM-myl7:GFP* allele. Precise 5' and 3' junctions were confirmed in heart *myl7:eGFP* expressing F2 fin clipped adults.

## Quantitative RT-PCR

Experiments to measure endogenous gene mRNA levels by Reverse Transcription-quantitative PCR were designed and performed according to MIQE and updated guidelines (**Bustin et al., 2009**; **Taylor et al., 2019**). Three biological replicates were performed, with each replicate representing embryos from a different mating pair of fish. At 3 dpf 20 randomly selected larvae from incrosses of heterozygous $rbbp4^{off/+}$, $rbbp4^{on/+}$, $rb1^{off/+}$, and $rb1^{on/+}$ were collected for RNA extraction and genotyping. Individual dissected head tissue was placed in RNA*later* (Qiagen/Thermo Fisher AM7020) or DNA/ RNA shield (Zymo Research R1100-50) and individual trunk tissue was placed in 50 mM NaOH for genotyping. Five heads of each genotype were pooled and total RNA extracted using the Direct-zol RNA Microprep kit (Zymo Research, R2060) and the quality determined using a Bioanalyzer 2100 (Agilent) at the Iowa State University DNA Facility. RNA samples with a RIN >5 were normalized to the same concentration, and first strand cDNA was synthesized using SuperScript III First-Strand Synthesis SuperMix (ThermoFisher, 11752050) containing random hexamer and oligo dT primers. Primers were designed to amplify ~200 bp amplicons with an annealing temperature of 60 °C. Primer optimization and validation was performed with 3 primer concentrations (100, 200, and 400 nM) and 3 cDNA amounts (5, 25, and 215 ng) with two replicates per condition. Primer efficiency was calculated as described (**Bustin et al., 2009**; **Taylor et al., 2019**) and primer pairs with 90–100% efficiency were used for qPCR of control and test samples. The sequence of $rbbp4$, $rb1$ and reference gene $rps6kb1b$ qPCR primers are listed in **Table 4**. qPCR was performed on each sample in triplicate using SsoAdvanced Universal SYBR Green Supermix (Bio-Rad, 1725270) on a CFX Connect Real-Time System (Bio-Rad).

## Cre and Dre-mediated inversion of UFlip alleles

For synthesis of *Cre* and *Dre* mRNAs, the expression vectors *pT3TS-Cre* (**Clark et al., 2011**) and *pT3TS-Dre* were linearized with *Sal*I (New England Biolabs, R0138S) and *BamH*I (New England Biolabs, R0136S), respectively. 1 µg linearized vector was purified with the PureYield Plasmid Miniprep System (Promega, A1223) and used as template for in vitro synthesis of capped mRNA with the Ambion mMessage Machine T3 Transcription Kit (Thermo Fisher, AM1348). In vitro synthesized mRNA was purified with the RNA Clean and Concentrator Kit RCC (Zymo, R1013). A total of 12.5 pg *Cre* or 15 pg *Dre* mRNA was injected into one-cell stage embryos to promote recombination-mediated inversion of the UFlip cassette at *lox* or *rox* sites. UFlip cassette inversion in *Cre* or *Dre* injected 3 dpf larvae was confirmed by digestion of individual larvae in 50 mM NaOH at 95 °C for 30 min and neutralization by addition of 1/10[th] volume 1 M Tris-HCl pH 8.0. Genomic DNA/UFlip cassette 5' and 3' junctions were amplified by PCR with gene specific and UFlip primers listed in **Table 4**, followed by direct Sanger sequencing.

## Zebrafish tissue embedding, sectioning, immunolocalization, and imaging

Zebrafish embryo and larvae fixation, embedding, sectioning, and immunolabeling was as described previously (**Schultz et al., 2018**). Zebrafish embryos and larvae were anesthetized in 160 µg/ml Ethyl 3-aminobenzoate methanesulfonate (Tricaine, MS-222) $C_9H_{11}NO_2 \cdot CH_4SO_3$ (Sigma-Aldrich, 886-86-2) in E3 embryo media (**Westerfield, 1995**) and head and trunk dissected. Trunk tissue was placed in 20 µl 50 mM NaOH for genotyping. Heads were fixed in 4% paraformaldehyde overnight at 4 °C, incubated in 30% sucrose overnight at 4 °C, then processed and embedded in Tissue-Tek OCT (Fisher, 4583). Tissues were sectioned at 14–16 µm on a Microm HM 550 cryostat. Antibodies used for labeling: rabbit polyclonal anti-phospho-Histone H3 PH3 1:1000 (Cell Signaling Technology; 9701); mouse monoclonal anti-phospho-Histone H3 (Ser10), clone 3H10 1:500 (Millipore 05–806); mouse monoclonal anti-HuC/D 1:500 (Invitrogen A-21271); mouse monoclonal Anti-Cre recombinase 1:250 (Millipore-Sigma MAB3120); rabbit polyclonal anti-Caspase-3a 1:500 (BD Biosciences 559565); Alexa-594 (Invitrogen A-11005) and Alexa-488 (Invitrogen A-11008) conjugated secondary antibodies 1:500. Tissues were counterstained with 5 µg/ml DAPI, mounted in Fluoro-Gel II containing DAPI (Electron Microscopy Sciences 17985–50) and imaged on a Zeiss LSM800 laser scanning confocal microscope.

For live imaging, some embryos and larvae were treated with 0.003% 1-phenyl 2-thiourea (Sigma, P7629) to inhibit pigment synthesis, anesthetized in Tricaine in embryo media (**Westerfield, 1995**) and mounted on slides in 1.2% low-melt agarose/embryo media/Tricaine. Fluorescence and bright

field imaging were performed on a Zeiss SteREO Discovery V12 microscope equipped with an X-Cite 120 W Metal Halide lamp (Excilitas Technologies, X-Cite 120Q). Images were captured with a Cannon Rebel T3 camera using EOS Utility software (Cannon). Bright field and fluorescence images were merged in Photoshop (Adobe). Live imaging was also performed on a Leica M165 FC stereomicroscope equipped with EL 6000 light source and K5 passive cooled sCMOS camera using Leica LAS X Multi Channel Acquisition software. Live imaging of larval retina was performed on a Zeiss LSM800 laser scanning confocal microscope.

## Quantification and statistical analyses

Quantification of proliferation and apoptosis was performed on three sections of immunolabeled head tissue for each individual, from three biological replicates of zebrafish embryos or larvae. Prism (GraphPad) software was used for two-tailed unpaired Student's t-test with mean ± s.e.m. statistical analyses and production of bar graphs.

## Acknowledgements

The authors thank Dr Raquel Espin (Iowa State University) for the KalTA4 cDNA, and Jon Luiken and Tanya Schwab for technical assistance.

## Additional information

### Competing interests

Wesley A Wierson: has competing interests with LifEngine and LifEngine Animal Health. Stephen C Ekker: Reviewing editor, eLife and has competing interests with LifEngine and LifEngine Animal Health. Karl J Clark: has competing interests with Recombinetics Inc, LifEngine and LifEngine Animal Health. Jeffrey J Essner, Maura McGrail: has competing interests with Recombinetics Inc, Immusoft Inc, LifEngine and LifEngine Animal Health. The other authors declare that no competing interests exist.

### Funding

| Funder | Grant reference number | Author |
| --- | --- | --- |
| NIH Office of the Director | R24 OD 020166 | Stephen C Ekker<br>Karl J Clark<br>Jeffrey J Essner<br>Maura McGrail |
| Conselho Nacional de Desenvolvimento Científico e Tecnológico | | Maira P Almeida |

The funders had no role in study design, data collection and interpretation, or the decision to submit the work for publication.

### Author contributions

Fang Liu, Conceptualization, Investigation, Methodology, Validation, Visualization, Writing – review and editing; Sekhar Kambakam, Formal analysis, Investigation, Methodology, Resources, Validation, Visualization, Writing – review and editing, Conceptualization; Maira P Almeida, Conceptualization, Formal analysis, Investigation, Methodology, Resources, Validation, Writing – original draft, Writing – review and editing; Zhitao Ming, Conceptualization, Formal analysis, Investigation, Methodology, Resources, Validation, Visualization; Jordan M Welker, Conceptualization, Investigation, Methodology, Resources, Writing – review and editing; Wesley A Wierson, Formal analysis, Investigation, Methodology, Validation, Writing – review and editing; Laura E Schultz-Rogers, Formal analysis, Investigation, Methodology, Resources, Validation, Visualization, Writing – review and editing; Stephen C Ekker, Jeffrey J Essner, Conceptualization, Funding acquisition, Methodology, Project administration, Resources, Supervision, Writing – review and editing; Karl J Clark, Conceptualization, Funding acquisition, Investigation, Methodology, Project administration, Resources, Supervision, Writing – review

and editing; Maura McGrail, Conceptualization, Formal analysis, Funding acquisition, Investigation, Methodology, Project administration, Resources, Supervision, Visualization, Writing – original draft, Writing – review and editing

### Author ORCIDs
Sekhar Kambakam  http://orcid.org/0000-0002-3331-3754
Stephen C Ekker  http://orcid.org/0000-0003-0726-4212
Karl J Clark  http://orcid.org/0000-0002-9637-0967
Jeffrey J Essner  http://orcid.org/0000-0001-8816-3848
Maura McGrail  http://orcid.org/0000-0001-9308-6189

### Ethics
Use of zebrafish for research in this study was performed according to the Guidelines for Ethical Conduct in the Care and Use of Animals (APA, 1986), and carried out in accordance with Iowa State University Animal Care and Use Committee IACUC-18-279 and IACUC-20-058 approved protocols. All methods involving zebrafish were in compliance with the American Veterinary Medical Association (2020), ARRIVE (Percie du Sert et al., 2020) and NIH guidelines for the humane use of animals in research.

### Decision letter and Author response
Decision letter https://doi.org/10.7554/eLife.71478.sa1
Author response https://doi.org/10.7554/eLife.71478.sa2

## Additional files

### Supplementary files
• Transparent reporting form
• Source data 1. Gel image files.

### Data availability
All data generated or analyzed during this study are included in the manuscript and supporting files. Source data files have been provided for Table 2, Figure 4, Figure 4 - figure supplement 1, Figure 5, Figure 5 - figure supplement 2, Figure 6, Figure 6 - figure supplement 2, Figure 7, Figure 7 - figure supplement 1, Figure 7 - figure supplement 2, Figure 8, Figure 9. DNA constructs reported in this study have been deposited at Addgene in the Jeffrey Essner lab list (Plasmid IDs: 173886, 173887, 173888, 173889, 173890, 173891, 180007, 180008, 180009, 180010, 180011, 18012). DNA constructs and transgenic zebrafish lines are available on request to M. McGrail.

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
