## [Editor Report]

This technical paper describes a novel strategy for conditional mutagenesis in zebrafish. It develops a simple approach to isolate mutagenic Cre/*lox* conditional alleles for tissue-specific gene inactivation. This work will be of great interest for the zebrafish community, advancing the exciting genetic manipulation tool box in this model organism.

---

## [Decision Letter]

**Decision letter after peer review:**

Thank you for submitting your article "Zebrafish Cre/*lox* regulated UFlip alleles generated by CRISPR/Cas targeted integration provide cell-type specific conditional gene inactivation" for consideration by *eLife*. Your article has been reviewed by two peer reviewers, and the evaluation has been overseen by a Reviewing Editor and Richard White as the Senior Editor. The reviewers have opted to remain anonymous.

The reviewers have discussed their reviews with one another, and with the Reviewing editor and Senior Editor. All recognise the promise and potential of your approach, but the reviewers have identified a number of significant concerns.

It was agreed that extensive revisions and new data, including additional controls and quantitation, would be needed in any revised version.

Essential revisions:

1) Show the possibility to switch the On-Off orientation in a tissue-specific manner; in particular, demonstrate the ability to turn genes on (see comments from Reviewer 2).

2) Show the possibility to trace cells with the inactive gene via fluorescent reporter activation.

3) Test the utility of imprecise insertion, as it is not clear why it's needed to insert the construct with HDR in an intron.

4) Provide a quantitative assessment of the efficiency of flipping.

5) Strengthen the validation of the null phenotypes described, and of the expression of targeted genes.

Full details of the requested revisions are given in the reviews below.

*Reviewer #1 (Recommendations for the authors):*

I have few major concerns that limit in my view the impact of the manuscript.

The first is that the ability of following mutagenized cells that have undergone the cassette rearrangements from the Gene-ON to Gene -OFF orientation via fluorescent tagging of the endogenous gene is not presented here. This is in my opinion the main interesting feature of this approach and unfortunately is not shown here. Instead the putative mutagenized cells are revealed only by Cre expression that may be transient and doesn't allow to follow the inversion events.

Secondly, the impact of inserting such a large cassette, containing multiple fluorescent reporter ORF and an extra promoter as a transgenesis marker, on the transcription of the gene targeted and the surrounding loci should be better assessed. Ideally this should be done via RNAseq strategies as this intronic insertion could potentially affect the transcriptional landscape of the targeted region. This analysis should also reveal the exact extent of WT transcript splicing events in the mutagenic configuration now generally estimated at less than 1%.

Finally the authors build their strategy on their previously published GeneWeld strategy, using short homology harms to aim at precise integration in the targeted locus. This, as they discuss, reduce the efficiency of their approach compared to previously published strategies including non homology based approaches. Giving that small INDELS in an intronic integration of a large cassette most likely have little or no consequences, it is not clear to me why the use of non homology directed integration should not be favored here. At the very least, this should be tested as they have all the tools in hands to do in their target loci by targeting the same construct without the short homology harms and comparing the efficiency they obtain.

I believe these three points would strengthen their conclusion and help scientists to choose the best strategy when designing their conditional KO strategy.

*Reviewer #2 (Recommendations for the authors):*

In addition to the issues noted in the public review, the authors should also address a few additional points:

Improved presentation of data.

The readability of the text and "scannability" of the figures could be improved. There are a probably unnecessarily large number of somewhat confusingly designated and not always clearly labeled figures, making more work than should be required for the reader to navigate and digest the key features of this manuscript. While I do appreciate that the authors have provided a very large amount of supporting data allowing the reader to "delve into the details," the authors should consider reorganizing (if possible reducing the number of) the figures, relabeling them for greater clarity and to make it easier for the reader to extract essential information.

Genetic compensation.

It would be useful for the authors to at least discuss or potentially address with data whether and to what extent compensation occurs in the context of their transgene "off" alleles, since transcriptional compensation of CRISPR mutants has been a substantial problem and if the authors methods demonstrably avoid this problem it would be another significant advantage of their method. The authors could potentially target a well characterized gene with CRISPR allele compensation and compare the effects of their targeted "off" insertion.

Potential usefulness of imprecise insertions.

Since imprecise insertions are more common than precise insertions, it would also be useful for the authors to at minimum discuss and if possible address with data using the different alleles they have generated whether or not imprecise insertions work as well as precise insertions for purposes of both disrupting (in "off" alleles) or allowing gene expression (in "on" alleles). Since these are intronic insertions and many of the imprecise insertions seem likely to leave the transgene largely intact, many of them may work just as well as precise insertions for their intended purposes.

Also,

The following line from page 4 last paragraph lines 123-124

"… gene off orientation knock down gene expression to>99% of wild type levels …" appears to say that it was knocked down less than 1% from WT levels as stated. It clearly is meant to say something closer to "gene expression was reduced BY >99% of wild type levels…"

This turn of phrase was used more than once in the manuscript a careful combing of the manuscript for this error is warranted.

---

## [Author Response]

Essential revisions:1) Show the possibility to switch the On-Off orientation in a tissue-specific manner; in particular, demonstrate the ability to turn genes on (see comments from Reviewer 2).

– Tissue-specific conditional inactivation: Using the conditional *rbbp4^on^* and *rb1^on^* alleles, we demonstrate using cell type specific proneural Cre drivers that each gene is required in the progenitor population during brain development. Conditional inactivation of *rbbp4^on^* to “off” with *ascl1b-2A-Cre* leads to apoptosis in the developing midbrain optic tectum (Figure 6, Figure 6—figure supplement 1 and 2). Conditional inactivation of *rb1^on^* to “off” with *neurod1-2A-Cre* leads to inappropriate cell cycle entry in the midbrain tectum and retina. (Figure 9, Figure 9—figure supplement 1).

– Tissue-specific conditional rescue: Using the conditional *rbbp4^off^* and *rb1^off^* alleles we demonstrate the ability to turn a gene “on” and rescue a loss of function phenotype. Conditional rescue of *rbbp4^off^* to “on” with *ascl1b-2A-Cre* lead to a reduction in apoptosis in the midbrain, although not significant (Figure 5). Conditional rescue of *rb1^off^* to on with *neurod1-2A-Cre* was clear and robust (Figure 8 O), suppressing the mutant phenotype with significant reduction in the number of mutant cells throughout the midbrain and retina.

– Ubiquitous conditional inactivation and rescue: By Cre injection, we demonstrate robust conditional inactivation (“on” to “off”) and induction of mutant phenotypes for both genes (Figures 6 and 9). We also demonstrate robust conditional activation (“off” to “on”) and rescue of mutant phenotypes for both genes (Figures 5 and 8).

– In all cases we provide rigorous quantification and statistical analysis of phenotypic changes and measure the frequency of Cre-mediated recombination.

2) Show the possibility to trace cells with the inactive gene via fluorescent reporter activation.

We corrected the mRFP gene trap in the UFlip vector by addition of the porcine virus teschovirus^-1^ 2A peptide sequence. We isolated an *rbbp4^off^* and *rbbp4^on^* allele with this updated UFlip-2A-mRFPvector. The *rbbp4^off^* allele shows strong mRFP expression in the expected tissues (Figure 4 A – H) that is lost after Cre recombination (Figure 5, Figure 5 —figure supplement 1). The *rbbp4^on^* allele does not express mRFP (Figure 4 I-N), as expected. Introduction of Cre leads to recombination of the gene trap cassette into the active orientation and induction of mRFP expression (Figure 6, Figure 6 —figure supplement 1).

3) Test the utility of imprecise insertion, as it is not clear why it's needed to insert the construct with HDR in an intron.

The *rb1^off^* allele used to establish an F3 generation has a 5’ junction with a homology arm duplication and 115 bp insertion (Figure 2C). This did not impact the function of the allele or the ability of Cre to recombine from “off” to “on”, leading to conditional rescue.

The value of using HDR is an increase in the frequency of on target integration, therefore fewer adults need to be screened to identify a transmitting founder.

4) Provide a quantitative assessment of the efficiency of flipping.

For each Cre recombination experiment that was performed (Figures 5, 6, 8, 9), we used quantitative PCR on genomic DNA to measure the loss of the 5’ and 3’ junctions of the original allele. Data plots are included in the main figure or in a figure supplement.

5) Strengthen the validation of the null phenotypes described, and of the expression of targeted genes.

For each conditional allele that was isolated we performed RT-qPCR on embryos from each genotypic class of a conditional allele incross, to determine the impact of the targeted allele on gene expression (Figures 4, 5).

Quantification of mutant phenotypes and statistical analyses to determine significance have been performed for 1. The characterization of the conditional alleles 2. The assessment of conditional rescue or inactivation. Data plots with *p* values are included in all figures and figure legends.

Full details of the requested revisions are given in the reviews below.Reviewer #1 (Recommendations for the authors):I have few major concerns that limit in my view the impact of the manuscript.The first is that the ability of following mutagenized cells that have undergone the cassette rearrangements from the Gene-ON to Gene -OFF orientation via fluorescent tagging of the endogenous gene is not presented here. This is in my opinion the main interesting feature of this approach and unfortunately is not shown here. Instead the putative mutagenized cells are revealed only by Cre expression that may be transient and doesn't allow to follow the inversion events.

We fully agree with the reviewer and thank them for requesting this critical improvement to the UFlip approach. As described above, we added a 2A peptide to our UFlip cassette and isolated new “on” and “off” alleles. These alleles show mRFP expression in the active, gene “off” orientation.

Secondly, the impact of inserting such a large cassette, containing multiple fluorescent reporter ORF and an extra promoter as a transgenesis marker, on the transcription of the gene targeted and the surrounding loci should be better assessed. Ideally this should be done via RNAseq strategies as this intronic insertion could potentially affect the transcriptional landscape of the targeted region. This analysis should also reveal the exact extent of WT transcript splicing events in the mutagenic configuration now generally estimated at less than 1%.

We used both RT-qPCR and phenotypic assessment to determine the impact of the cassette integration on gene expression and gene activity. As suggested, we included RT-qPCR results for the wildtype exon-exon splicing that would be disrupted by the cassette integration in the intron. We also examined splicing between exons downstream of the integration. These results demonstrate that in the gene “off” orientation endogenous gene expression is 99% knocked down. Integration in the passive “on” orientation did reduce expression in homozygotes (*rbbp4^on/on^* Figure 4 J reduced by 40.7%; *rb1^on/on^* Figure 7 J 1 reduced by 17.1%). However, the reduction in expression did not lead to mutant phenotype in heterozygotes ,or in combination with loss of function alleles (Figure 4 K-N; Figure 7, K-P), indicating gene activity was not disrupted. This may be gene dependent.

Finally the authors build their strategy on their previously published GeneWeld strategy, using short homology harms to aim at precise integration in the targeted locus. This, as they discuss, reduce the efficiency of their approach compared to previously published strategies including non homology based approaches. Giving that small INDELS in an intronic integration of a large cassette most likely have little or no consequences, it is not clear to me why the use of non homology directed integration should not be favored here. At the very least, this should be tested as they have all the tools in hands to do in their target loci by targeting the same construct without the short homology harms and comparing the efficiency they obtain.

The *rb1^off^* allele used for the experiments in Figures 7 and 8 has a 5’ junction with a homology arm duplication and 115 bp insertion (Figure 2C). This did not impact the function of the allele or the ability of Cre to recombine the allele from “off” to “on”, leading to conditional rescue.

I believe these three points would strengthen their conclusion and help scientists to choose the best strategy when designing their conditional KO strategy.

Thank you again for your critical suggestions that have vastly improved the UFlip approach to generate conditional alleles, and our manuscript.

Reviewer #2 (Recommendations for the authors):In addition to the issues noted in the public review, the authors should also address a few additional points:

Thank for your critical suggestions that have vastly improved the UFlip approach to generate conditional alleles, and our manuscript.

Improved presentation of data.The readability of the text and "scannability" of the figures could be improved. There are a probably unnecessarily large number of somewhat confusingly designated and not always clearly labeled figures, making more work than should be required for the reader to navigate and digest the key features of this manuscript. While I do appreciate that the authors have provided a very large amount of supporting data allowing the reader to "delve into the details," the authors should consider reorganizing (if possible reducing the number of) the figures, relabeling them for greater clarity and to make it easier for the reader to extract essential information.

Thank you for this suggestion. We simplified the allele designation and reorganized the manuscript substantially from the first revision. By focusing on just two genes, and repeating the same analyses for the “on” and “off” alleles for both genes, we hope this improves readability and overall “flow”. We also hope the overall presentation of the data will make it easier for the reader to grasp the major points of the manuscript.

Genetic compensation.It would be useful for the authors to at least discuss or potentially address with data whether and to what extent compensation occurs in the context of their transgene "off" alleles, since transcriptional compensation of CRISPR mutants has been a substantial problem and if the authors methods demonstrably avoid this problem it would be another significant advantage of their method. The authors could potentially target a well characterized gene with CRISPR allele compensation and compare the effects of their targeted "off" insertion.

This is an important topic that our approach could address. But it is well beyond the scope of our current study. Indeed, we intentionally chose essential genes with robust, well-defined loss of function phenotypes that we previously described using stable indel alleles. In our opinion, our demonstration of endogenous conditional gene inactivation, and rescue, is highly impactful and stands alone as an important step forward for the zebrafish community.

Potential usefulness of imprecise insertions.Since imprecise insertions are more common than precise insertions, it would also be useful for the authors to at minimum discuss and if possible address with data using the different alleles they have generated whether or not imprecise insertions work as well as precise insertions for purposes of both disrupting (in "off" alleles) or allowing gene expression (in "on" alleles). Since these are intronic insertions and many of the imprecise insertions seem likely to leave the transgene largely intact, many of them may work just as well as precise insertions for their intended purposes.

As described in the response to essential revision Point 3., we used both precise alleles and an imprecise allele (5’ homology arm duplication plus 155bp insertion). The *rb1^off^* allele used to establish an F3 generation has a 5’ junction with a homology arm duplication and 115 bp insertion (Figure 2C). This did not impact the function of the allele or the ability of Cre to recombine from “off” to “on”, leading to conditional rescue.

The value of using HDR is an increase in the frequency of on target integration with both sides intact, therefore fewer adults need to be screened to identify a transmitting founder. In other words, we identified founders transmitting alleles at a rate of 9-30%. However, half or more of those alleles are missing either a 5’ or 3’ junction. So the rate of recovery of an on target integration with both a 5’ and 3’ junction, drops to 4-14%. Preselection of gene “off” injected embryos showing the gene trap reporter 2A-mRFP expression appeared to increase the efficiency of recovering an on target, precise allele (12.5 – 14% Table 1) vs. recovery of a gene “on” allele where there isn’t a marker to preselect for on target integration (*rb1^on is57^* 4% (1/25 adults screened)).

Also,The following line from page 4 last paragraph lines 123-124"… gene off orientation knock down gene expression to>99% of wild type levels …" appears to say that it was knocked down less than 1% from WT levels as stated. It clearly is meant to say something closer to "gene expression was reduced BY >99% of wild type levels…"This turn of phrase was used more than once in the manuscript a careful combing of the manuscript for this error is warranted.

Thank you for catching this. The manuscript text has been corrected.